# *In vivo* gastrointestinal drug-release monitoring through second near-infrared window fluorescent bioimaging with orally delivered microcarriers

Rui Wang[1], Lei Zhou[1], Wenxing Wang[1], Xiaomin Li[1] & Fan Zhang[1]

Non-invasive monitoring of gastrointestinal drug release *in vivo* is extremely challenging because of the limited spatial resolution and long scanning time of existing bioimaging modalities, such as X-ray radiation and magnetic resonance. Here, we report a novel microcarrier that can retain drugs and withstand the harsh conditions of gastrointestinal tract. Significantly, we can track the microcarrier fate and semi-quantitatively monitor the content of drug released *in vivo* in real time by measuring the fluorescence signals in the second near-infrared window of lanthanide-based downconversion nanoparticles with an absorption competition-induced emission bioimaging system. The microcarriers show a prolonged residence time of up to 72 h in the gastrointestinal tract, releasing up to 62% of their content. Moreover, minimal deposition of the microcarriers is found in non-target organs, such as the liver, spleen and kidney. These findings provide novel insights for the development of therapeutic and bioimaging strategies of orally administered drugs.

[1] Department of Chemistry, Laboratory of Advanced Materials, Collaborative Innovation Center of Chemistry for Energy Materials, State Key Laboratory of Molecular Engineering of Polymers, Fudan University, Shanghai 200433, China. Correspondence and requests for materials should be addressed to F.Z. (email: zhang_fan@fudan.edu.cn).

Oral administration is by far the most convenient pathway for drug delivery because of its non-invasive nature, avoiding the pain and discomfort associated with injections as well as minimizing contaminations[1–3]. Despite these advantages, most bioactive drugs, such as peptides and proteins, suffer from low bioavailability after oral administration because of their degradation or deactivation in harsh gastrointestinal (GI) environments[4,5]. Compared with traditional therapeutic agents, microcarrier drug delivery systems including polymeric nanoparticles[6–8], liposome particles[9,10], nanoemulsions[11] and mesoporous silica particles (MSP)[12–14], offer many attractive features, such as high loading capacities, efficient protection of bioactive drugs and high therapeutic indexes[15–17]. However, clinical translation can only be realized if the *in vivo* delivery of the microcarriers is perfectly evaluated, including the biodistribution, pharmacokinetics, drug-release behaviour, clearance route and toxicity.

To evaluate the delivery of the microcarrier and monitor the drug-release process in real time, biomedical imaging techniques such as position-emission tomography (PET)[18], photoacoustic imaging[19] and magnetic resonance imaging (MRI)[20] have been utilized. However, the key constraints of these tomographic techniques are their limited spatial resolution and long image imaging times, making them incapable of visualizing the real-time biodistribution and pharmacokinetics of microcarriers. In comparison, fluorescent bioimaging is a real-time, non-invasive and radiation-free technology[21,22]. Nonetheless, most of the fluorophores emit only in the visible range (400–750 nm) and first near-infrared (NIR-I) window (750–900 nm) with a low tissue penetration depth of several micrometres to millimetres because of the severe absorption and scattering of photons by tissues[23,24]. Recently, fluorescent agents such as quantum dots (QDs)[25,26], single-walled carbon nanotubes (SWNTs)[27,28] and lanthanide-based downconversion nanoparticles (DCNPs)[23,29] emitting in the second near-infrared (NIR-II) window (1,000–1,400 nm) have been widely reported. These NIR-II imaging agents feature deeper penetration depths (1–2 cm)[29] and higher resolution (sub-10 μm)[30] with reduced scattering and minimized autofluorescence[24,27,28,31]. Especially, DCNPs play a very important role in NIR-II fluorescent bioimaging applications due to their distinct properties, such as non-photobleaching activity, long lifetimes, high efficiency and multiexcitation capabilities[23,32,33].

In this work, we report a novel type of DCNP-based NIR-II fluorescent mesoporous microcarrier for orally delivering protein drugs. The NIR-II bioimaging results demonstrate that these protein drugs showed little leakage in the stomach (pH = 1) and duodenum (pH = 5) but sustained release in the intestine (pH = 8). Moreover, the microcarriers exhibited a prolonged residence time of up to 72 h in the intestine, with little deposition (< 0.1%) in other organs such as the liver, spleen and kidney. We also designed an absorption competition-induced emission (ACIE) bioimaging system to track the microcarrier fate and semi-quantitatively monitor the drug-release percentage *in vivo* by simply switching the excitation source (Fig. 1a). We found that the GI effective protein-release percentage of the microcarriers reached ∼62% after 72 h of release. Moreover, the activity of the released enzymes from the orally delivered microcarriers was largely preserved, further indicating the feasibility of using microcarriers for oral drug delivery.

## Results

**Synthesis of NIR-II fluorescent mesoporous microcarriers.** The synthesis procedure for the NIR-II fluorescent mesoporous SiO₂-Nd@SiO₂@mSiO₂-NH₂@SSPI microcarriers (SSPI is short for succinylated soy protein isolate polymer) consisted of five

steps (Fig. 1b). Solid SiO₂ particles were fabricated via the traditional Stöber method[34]. The resulting solid SiO₂ particles showed a uniform size of 500 nm with a smooth surface (Fig. 2a) and highly negative zeta potential (− 38.4 mV) (Supplementary Fig. 1). NaGdF₄:5%Nd@NaGdF₄ DCNPs were prepared by a successive layer-by-layer (SLBL) strategy[35,36] (Supplementary Fig. 2a), followed by hydrophilic modification with amphiphilic 1,2-distearoyl-sn-glycero-3-phosphoethanolamine-N-[amino(polyethylene glycol)-2000] (DSPE-PEG2000-NH₂) phospholipids[37]. These hydrophilic DCNPs showed excellent monodispersity (∼14 nm) (Supplementary Fig. 2b) and a positive zeta potential (+ 5 mV). After simply mixing the DCNPs with SiO₂ particles, the obtained hydrophilic DCNPs were easily attached on the surface of the SiO₂ particles to form SiO₂-Nd nanoparticles through electrostatic interactions (Fig. 2b,c; Supplementary Fig. 1b). According to the inductively coupled plasma atomic emission spectrometry (ICP) measurements, ∼200 DCNPs were attached on each of the SiO₂ particle. After being coated with another layer of SiO₂ by the Stöber method, these DCNPs were firmly fixed in the framework of the SiO₂ particles (Fig. 2d). Then, the SiO₂-Nd@SiO₂ nanoparticles (NPs) were further coated with mesoporous SiO₂ shells using a biphase stratification method to obtain monodispersed SiO₂-Nd@SiO₂@mSiO₂ with a uniform size of ∼900 nm (Fig. 2e,f)[14]. The DCNPs attached in the SiO₂ core could still be clearly identified in high-resolution transmission electron microscope (HRTEM) image (Fig. 2f, inset). The amino functionalization of SiO₂-Nd@SiO₂@mSiO₂ was successfully achieved via a post-grafting approach after extracting the surfactant, finally endowing these particles with a positive charge (+ 12.4 mV) (Supplementary Fig. 1). Figure 2g shows that the opening channels were retained in the resulting SiO₂-Nd@SiO₂@mSiO₂-NH₂ NPs. The BET surface area was measured to be ∼355 and ∼279 m² g⁻¹ before and after the modification of amino groups, respectively. Using the Barrett–Joyner–Halenda (BJH) model, the total pore volume and average pore size of the samples before and after surface functionalization were estimated to be ∼0.70 cm³ g⁻¹, ∼9.8 nm and ∼0.50 cm³ g⁻¹, ∼8.6 nm, respectively (Supplementary Fig. 3a,b). In addition, the small-angle X-ray scattering (SAXS) patterns showed a constant single scattering peak at ∼0.63 nm⁻¹ before and after the amino functionalization, suggesting that the uniform mesostructures were well retained during the surface modification process (Supplementary Fig. 3c). The SiO₂-Nd@SiO₂@mSiO₂-NH₂ NPs were encapsulated within a smart succinylated soy protein isolate (SSPI) polymer, which has been proven to be a good stimuli-responsive material for intestinal release systems due to its biosafety and sensitive responsiveness to pH[38]. After SSPI grafting, the outer surface of the microcarriers became rough (Fig. 2h,i), and the inner DCNPs were still clearly recognized in the core/shell structured microcarriers through high-angle annular dark-field scanning transmission electron microscopy (HAADF-STEM) (Supplementary Fig. 4). Elemental mapping of single SiO₂-Nd@SiO₂@mSiO₂-NH₂@SSPI particle further demonstrated its core/shell geometry and composition, including Si in the SiO₂ core and mSiO₂ shell, Gd in the DCNPs and C in the SSPI (Fig. 2j).

**In vitro GI protein-release monitoring with the ACIE system.** To realize quantitative, real-time monitoring of the drug release, an ACIE bioimaging method was fabricated. To select proper absorbing competition acceptors for the ACIE method, several criteria must be followed: absorption overlap with only one of the multiexcitation bands of the DCNPs having a higher molar absorption coefficient, a small molecular size to enter or exit the mesopores freely, no emission in the NIR-II region and a strong affinity for co-loaded protein drugs. On the basis of the above criteria, the NPTAT organic dye with a maximum absorption at

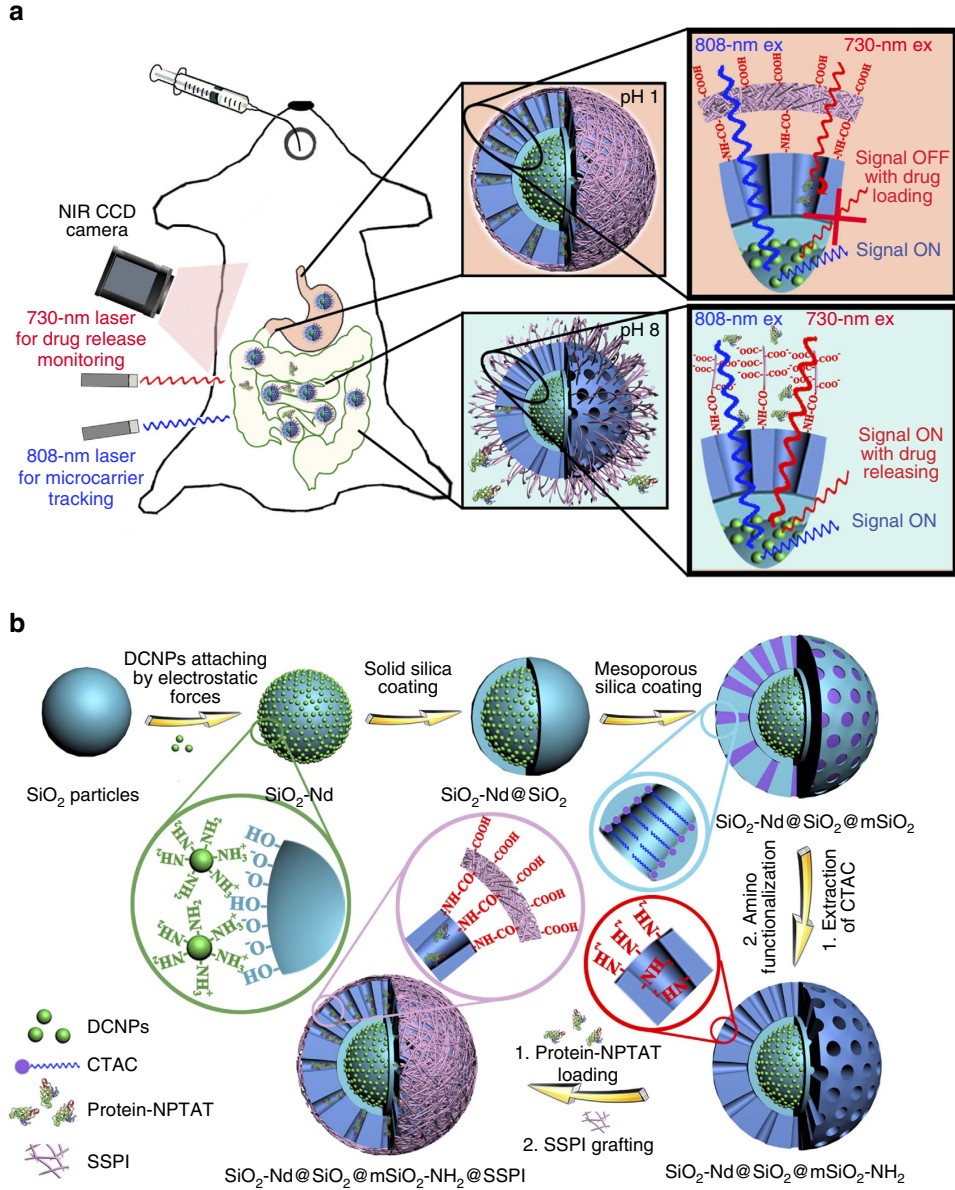

**Figure 1 | Schematic illustration of the ACIE bioimaging system and fabrication procedure.** (**a**) Microcarrier fate tracking and drug-release monitoring by the ACIE bioimaging system using an InGaAs CCD camera. After oral administration, the BSA–NPTAT-loaded microcarriers showed little protein drugs leakage in the GI tract (pH = 1) but sustained release in the intestine (pH = 8) due to the deprotonation of the SSPI on the outer surface of the microcarrier. (**b**) Procedures of NIR-II fluorescent mesoporous microcarriers preparation, protein drugs loading and SSPI grafting.

625 nm was chosen as the absorbing competition acceptor for the DCNPs[39]. As shown in Fig. 3a and Supplementary Fig. 5, the absorption coefficient of the NPTAT was ∼2,000-fold higher than that of the NaGdF$_4$:5%Nd@NaGdF$_4$ DCNPs at 730 nm, while it barely absorbed 808-nm light. Moreover, no fluorescence (FL) signal from the NPTAT interfered with the NIR-II signals of the DCNPs under 730- or 808-nm excitation. As shown in Fig. 3b, the NIR-II signals of the DCNPs at 1,060 nm under 730-nm excitation were quenched accordingly with the increase in NPTAT loading in the microcarriers. The quenching efficiency of the NIR-II signals excited by 730-nm excitation reached 100% when the NPTAT loading amount was increased to 0.19 wt.%; meanwhile, <10% quenching was observed during 808-nm excitation (Supplementary Fig. 6). Significantly, there was a positive correlation between the loaded amount of NPTAT and the quenching efficiency (Fig. 3c). These interesting characteristics provided an opportunity to monitor the drug-

release process with NIR-II bioimaging under 730-nm excitation while tracking the fate of the microcarriers under 808-nm excitation. Moreover, the NIR-II signals under 808-nm excitation were also used as a reference to estimate the release percentage in real time because of their inconspicuous response to the NPTAT.

The interaction between phthalocyanines (including NPTAT) and proteins has been widely investigated using FL spectroscopy methods[40,41]. By evaluating the FL quenching effect of proteins after adding a certain amount of NPTAT, the binding constant ($K_A$) and bimolecular quenching constant ($k_q$) between them can be quantified by the Stern–Volmer equation (see the 'Methods' section, equations (3–5)). The variable $K_A$ reflects the degree of interaction between the NPTAT and proteins, while $k_q$ can be used to identify the type of quenching process (static or dynamic quenching)[40]. In the present work, the $k_q$ for the BSA–NPTAT complex was calculated to be $7.2 \times 10^{12} \, \mathrm{M}^{-1} \mathrm{s}^{-1}$, which is much higher than the normal value for dynamic quenching

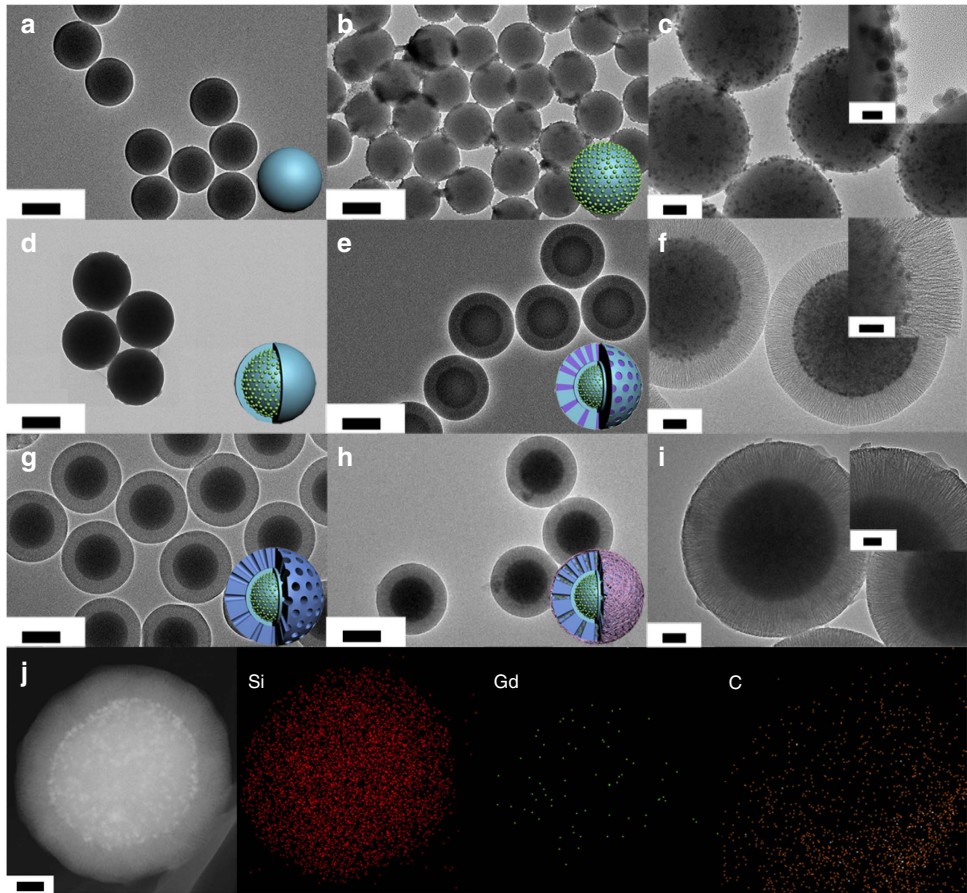

**Figure 2 | Structural characterization of the as-synthesized microcarriers.** (**a**–**i**) TEM images of solid SiO$_2$ particles (**a**), SiO$_2$-Nd (**b**,**c**), SiO$_2$-Nd@SiO$_2$ (**d**), SiO$_2$-Nd@SiO$_2$@mSiO$_2$ (**e**,**f**), SiO$_2$-Nd@SiO$_2$@mSiO$_2$-NH$_2$ (**g**) and SiO$_2$-Nd@SiO$_2$@mSiO$_2$-NH$_2$@SSPI (**h**,**i**). Inset images in **c**,**f**,**i** are the corresponding HRTEM images. (**j**) Dark-field STEM image of a single SiO$_2$-Nd@SiO$_2$@mSiO$_2$-NH$_2$@SSPI microcarrier and EDXS element mapping of elements Si, Gd and C in one microcarrier. (**a**,**b**,**d**,**e**,**g**,**h**) Scale bars, 500 nm; (**c**,**f**,**i**,**j**) scale bars, 100 nm; (**c**) scale bars, inset image 20 nm; and (**f**,**i**) scale bars, in the inset images 50 nm.

(*ca.* $10^{10}\,\mathrm{M}^{-1}\,\mathrm{s}^{-1}$) (Supplementary Table 1), confirming that the quenching process between NPTAT and BSA belongs to static quenching induced by strong interactions (Supplementary Fig. 7). Various BSA–NPTAT complexes with different molar ratios (Supplementary Fig. 7a) can be used during BSA release monitoring as long as the corresponding NPTAT content in the microcarrier is kept at 0.19 wt.% to ensure a maximum quenching effect. On the basis of these principles, the weight ratio of the BSA and NPTAT in the BSA–NPTAT complex was fixed at 500:1 (equal to a 7.4:1 molar ratio, Supplementary Fig. 7) in the following tests.

As a proof of concept, BSA–NPTAT was loaded into the mesopores of the microcarriers with a capacity of 23.1 wt.% (the maximum loading capacity for BSA is 47.4 wt.%). Optical images showed that the microcarriers turned blue after BSA–NPTAT loading (Fig. 3d, inset). The *in vitro* drug-release behaviour of the BSA–NPTAT-loaded microcarriers was monitored in simulated GI tract fluids with different pH values (pH = 1, 5 and 8). As shown in the release profiles (Fig. 3d; Table 1), there was no noticeable protein releases at pH 1 and 5, while a burst release of BSA–NPTAT was triggered at pH 8, followed by sustained release behaviour. An explanation of this phenomenon is that the SSPI showed insoluble features at pH 1 and 5 due to the protonation of its abundant carboxylic groups, while these carboxylic groups were largely deprotonated at pH 8, resulting in higher solubility in water to trigger the drug release from the mesopores (Fig. 1b). To investigate the feasibility of drug-release monitoring using NIR-II

signals from the microcarriers, we compared three groups of the measurement results: the released amount of BSA, monitored by FL spectroscopy; the released amount of NPTAT, monitored by ultraviolet–vis spectroscopy and the recovered amount of the NIR-II FL intensity of the microcarriers, monitored by NIR FL spectroscopy under 730-nm excitation. Figure 3d demonstrates that these results have a good correlation, suggesting that the ACIE method was sufficient for drug-release monitoring. Moreover, as shown in Table 1, the release rate of the BSA was always ∼500 times higher than that of the NPTAT, which was exactly equal to the mass ratio of BSA and NPTAT in the microcarriers (500:1), indicating that NPTAT can be utilized as an excellent tracer for BSA. To verify the feasibility of real-time protein-release monitoring through NIR bioimaging under 730-nm excitation, *in vitro* protein release at different times was monitored using an InGaAs camera. The NIR-II signals were enhanced gradually as the releasing time was extended, while the blue colour of the microcarriers faded gradually (Fig. 3e,f). Significantly, the release behaviour monitored by the NIR-II bioimaging method was also well correlated with the spectroscopy results (Fig. 3d), indicating that the ACIE method is a feasible approach to monitor real-time drug-release.

**In vivo biodistribution of the microcarriers and BSA.** To investigate the *in vivo* biodistribution properties of the microcarriers, different sized microcarriers (25, 80, 300 and 1,000 nm) were prepared followed by SSPI surface grafting

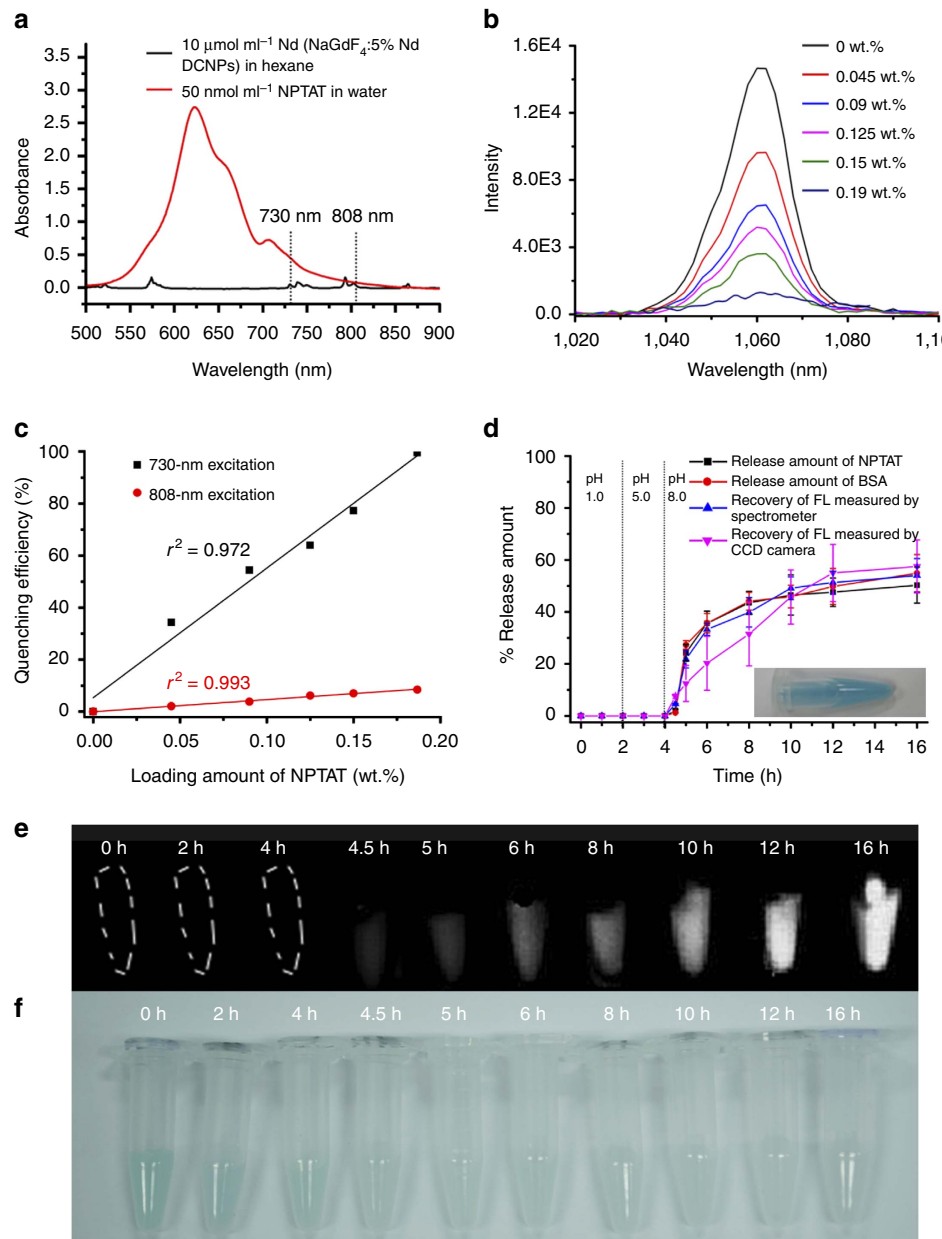

**Figure 3 | Feasibility investigation of the ACIE method for *in vitro* drug-release monitoring. (a)** Absorption spectra of 10 μmol ml$^{-1}$ DCNPs dispersed in hexane and 50 nmol ml$^{-1}$ NPTAT dispersed in water. It should be noted that the concentration of the DCNPs was determined from the Nd element content in the solvent. **(b)** NIR FL signals as a function of the NPTAT loading amounts in the microcarrier under 730-nm excitation. **(c)** NIR FL intensity of microcarriers loaded with different amounts of NPTAT under 730- or 808-nm excitation, respectively. **(d)** Time-dependent release profiles of BSA–NPTAT-loaded microcarriers in simulated GI tract fluids. Inset shows the optical graph of the NPTAT-loaded microcarriers (mean ± s.d. for $n = 3$). **(e,f)** NIR FL and optical images of microcarriers at different release time points. Representative images are for $n = 3$ per group.

(Supplementary Fig. 8). Five groups of mice were gavaged with 25, 80, 300 and 1,000 nm particles, and saline water as the positive groups and negative control, respectively. Then, NIR-II images at different time points (0.25, 2, 4, 6, 12, 24, 48 and 72 h) were taken for each group upon 808-nm irradiation (0.2 W cm$^{-2}$) (Fig. 4a; Supplementary Fig. 9). The *in vivo* collected NIR-II signals from the 25-nm group vanished in 12 h after gavaging, while those from the 300-nm group lasted 48 h. Significantly, highly resolved NIR-II signals were still observed at 72 h for the 1,000-nm group, indicating that larger particles can remain longer in the GI tract. This finding was also confirmed by the time-dependent distributions of the particles in different visceral organs measured by ICP (Fig. 4b–g). More than 90% of the 25-nm particles and 75%

of the 80-nm particles had cleared out of the bodies within 24 h. In contrast, only 55% of the 300-nm particles and 40% of the 1,000-nm particles had cleared at 24 h, and ∼20% of the 1,000-nm particles remained in the intestine at 72 h. As shown in Fig. 4b–g, the uptake extent of the particles through the GI tract was size dependent. The uptake of the 25-, 80- and 300-nm microcarriers showed a steady increase after oral administration, followed by a gradual decrease. The highest uptake amount (including the liver, spleen and kidney) was observed at 12 h after gavaging (1.15, 0.58 and 0.41% for the 25-, 80- and 300-nm microcarriers, respectively). In contrast, the uptake amount of the 1,000-nm microcarriers was below 0.1% throughout the oral drug delivery process (Supplementary Fig. 10), indicating that a minor

**Table 1 | *In vitro* release rates (*dc/dt*) of BSA and NPTAT.**

| Time (h) | BSA (mg ml$^{-1}$ h$^{-1}$) | NPTAT (µg ml$^{-1}$ h$^{-1}$) | $r_{BSA}/r_{NPTAT}$ |
|---|---|---|---|
| 0 | 0.0 | 0.0 | — |
| 1 | 0.0 | 0.0 | — |
| 2 | 0.0 | 0.0 | — |
| 3 | 0.0 | 0.0 | — |
| 4 | 0.0 | 0.0 | — |
| 4.5 | 22.4 | 47.8 | 470 |
| 5 | 13.4 | 28.1 | 477 |
| 6 | 4.8 | 9.7 | 495 |
| 8 | 0.6 | 1.2 | 500 |
| 10 | 8.0E − 02 | 1.4E − 01 | 557 |
| 12 | 1.0E − 02 | 2.0E − 02 | 500 |
| 16 | 0.0 | 0.0 | — |

health risk would be caused by the 1,000-nm microcarriers. Furthermore, by comparing the results measured using the ICP and NIR bioimaging techniques, we found that the differences between them were not significant ($P > 0.05$) (Fig. 5a), demonstrating that the NIR-II bioimaging approach is rather convincing for *in vivo* monitoring of the biodistribution of the microcarriers.

In addition to the biodistribution of the microcarriers, the pharmacokinetic behaviour of the released BSA was also investigated using a radioisotopic tracing method. As a proof of concept, [125]I-BSA-loaded 1,000-nm microcarriers were prepared and orally administered to mice. The radioisotopic tracing results from plasma, organs (liver, kidney, stomach and intestine) and excretion (including urine and excrement) showed that part of the released BSA was absorbed by the GI tract into the bloodstream, reaching nearly 1.7% D g$^{-1}$ (% D g$^{-1}$ is short for percentage of dose per gram) in the plasma 6 h after oral administration, while the rest of the absorbed BSA accumulated in the visceral organs such as the liver and kidney ($\sim 2\%$ D g$^{-1}$ for the liver and $\sim 3\%$ D g$^{-1}$ for the kidney). After 72 h of circulation, $\sim 25\%$ of the BSA had cleared out of the body through the urine (Fig. 5b–d).

***In vivo* GI drug-release monitoring through NIR-II bioimaging**. As a proof of concept, the NIR-II bioimaging technique was used to monitor the GI drug-release at different time points after gavaging BSA–NPTAT-loaded 1,000-nm microcarriers orally. A 730-nm and 808-nm laser was used as the excitation source to monitor the drug release and microcarrier fate, respectively (Fig. 6a,b). From the NIR bioimaging results under 808-nm excitation, it was clearly identified that most of the microcarriers were located in the stomach during the first 2 h and then moved to the intestine gradually. At 6 h after gavaging, the intensity of the NIR-II signals from the intestine reached a saturated level, followed by a continuous decrease due to clearance from the body. In contrast, the intensity of the NIR-II signals from the drug-loaded microcarriers under 730-nm excitation was slowly enhanced during the first 12 h and then decreased gradually. It was clearly demonstrated that the intensity of the NIR-II signals excited by the 808-nm laser was related to the quantity of the microcarriers in real time, while the intensity of the signals excited by the 730-nm laser reflected the drug-release percentage. Therefore, a semi-quantitative analysis was conducted to demonstrate the drug-release percentage ($Q_{(\tau)}$) of the microcarriers in real time:

$$Q_{(\tau)} = \frac{F_{730,(\tau)} - F_{730,0}}{F_{730,unloaded} - F_{730,0}} \times 100\%$$

$$= \frac{F_{730,(\tau)} - F_{730,0}}{\alpha F_{808,(\tau)} - F_{730,0}} \times 100\% \qquad (1)$$

where $F_{730,0}$ is the initial intensity under 730-nm excitation of the NIR-II signals from the drug-loaded microcarriers just after gavaging, and $F_{730,(\tau)}$ is used to represent the intensity of the NIR-II signals from the drug-loaded microcarriers under 730-nm excitation at a specific time point ($\tau$). The variable $F_{730,unloaded}$ is used to represent the intensity of the NIR-II signals from the unloaded microcarriers under 730-nm excitation. Because the intensity of the NIR-II signals under 730-nm excitation was quenched gradually with the increase of the drug loading amount, $F_{730,unloaded}$ was used to reflect the total amount of loaded BSA. However, it was difficult to obtain $F_{730,unloaded}$ in real time because of the quenching effect of the loaded drug. However, $F_{730,unloaded}$ can be related to $F_{808,(\tau)}$ directly because the NIR-II signals from the DCNPs under 730-nm and 808-nm excitation remained at a fixed ratio when the power density was below 4.55 W cm$^{-2}$ (Supplementary Figs 11 and 12; Supplementary Table 3). Therefore, $F_{730,unloaded}$ was expressed as $\alpha F_{808,(\tau)}$, where the coefficient $\alpha$ is the constant to represent $F_{730,unloaded}/F_{808,(\tau)}$, which was measured to be 0.62 (Supplementary Fig. 11). Furthermore, because the microcarriers are cleared out of the body with time, $F_{808,(\tau)}$ changed gradually during the drug delivery process. Therefore, $F_{808,(\tau)}$ was used to reflect the amount of unloaded microcarriers in real time. Taking the drug-release amount calculated at 12 h after gavaging as an example, the intensity of the NIR-II signals under 730-nm ($F_{730,(12 h)}$) or 808-nm ($F_{808,(12 h)}$) excitation was calculated to be 23.5 and 75.2, respectively (Fig. 6a). In addition, the $F_{730,0}$ was measured to be 3, so the release percentage $Q_{(12 h)}$ was estimated to be 47%. As shown in Fig. 6c, the calculated $Q$ was only 10% during the first 2 h, then showed a burst enhancement during the following 12 h and reached $\sim 62\%$ at 72 h, suggesting the efficient release behaviour of the microcarriers *in vivo*. We also investigated the *in vivo* drug-release kinetics by fitting the obtained $Q_{(\tau)}$ data to Supplementary Equation (S1) (Supplementary Fig. 13). As shown in Supplementary Fig. 13, the release profile behaved as a one-order release system with a high correlation ($r^2 > 0.92$), indicating the validity of the $Q_{(\tau)}$ values calculated by equation (1).

On the basis of the obtained release percentage $Q_{(\tau)}$, the cumulative release amount of the BSA in real time was derived from the following equation:

$$M_{(\tau)} = M_{(\tau-1)} + \frac{F_{808,(\tau-1)} + F_{808,(\tau)}}{2F_{808,0}} m\theta\left(Q_{(\tau)} - Q_{(\tau-1)}\right), \quad (2)$$

where $M_{(\tau)}$ and $M_{(\tau-1)}$ are the cumulative release amounts of BSA at two closed time points ($\tau$ and $\tau - 1$, respectively). The quantity $Q_{(\tau)} - Q_{(\tau-1)}$ reflects the release percentage during this time period (from time ($\tau - 1$) to time ($\tau$)). The variables $F_{808,(\tau-1)}$, $F_{808,(\tau)}$ and $F_{808,0}$ were used to represent the intensity of the NIR-II signals under 808-nm excitation from the drug-loaded microcarriers at time ($\tau - 1$), time ($\tau$) and the initial time point just after gavaging. Since the *in vivo* delivery of microcarriers is a dynamic process, the microcarriers are cleared out gradually, the real-time microcarrier percentage in the body can be calculated from $\frac{F_{808,(\tau-1)} + F_{808,(\tau)}}{2F_{808,0}}$. Coefficients $m$ and $\theta$ are the mass and drug loading percentage of the initial microcarriers, respectively. Therefore, the release amount of BSA from the microcarriers during a specific period (from time ($\tau - 1$) to time ($\tau$)) can be calculated from $\frac{F_{808,(\tau-1)} + F_{808,(\tau)}}{2F_{808,0}} m\theta\left(Q_{(\tau)} - Q_{(\tau-1)}\right)$. Taking the cumulative drug-release calculation from 6 to 12 h as an example, the $Q_{(12 h)}$ and $Q_{(6 h)}$ values were calculated to be 48 and 42% according to equation (1) (Fig. 6c). Meanwhile, the $F_{808,(6 h)}$, $F_{808,(12 h)}$ and $F_{808,0}$ values were measured to be 83.5, 75.2 and 93.3, respectively (derived from Fig. 6b), so the $M_{(12 h)}$ was estimated to be 103.3 ± 5.7 µg. The calculated cumulative drug-release amounts are shown in Table 2. It was shown that the

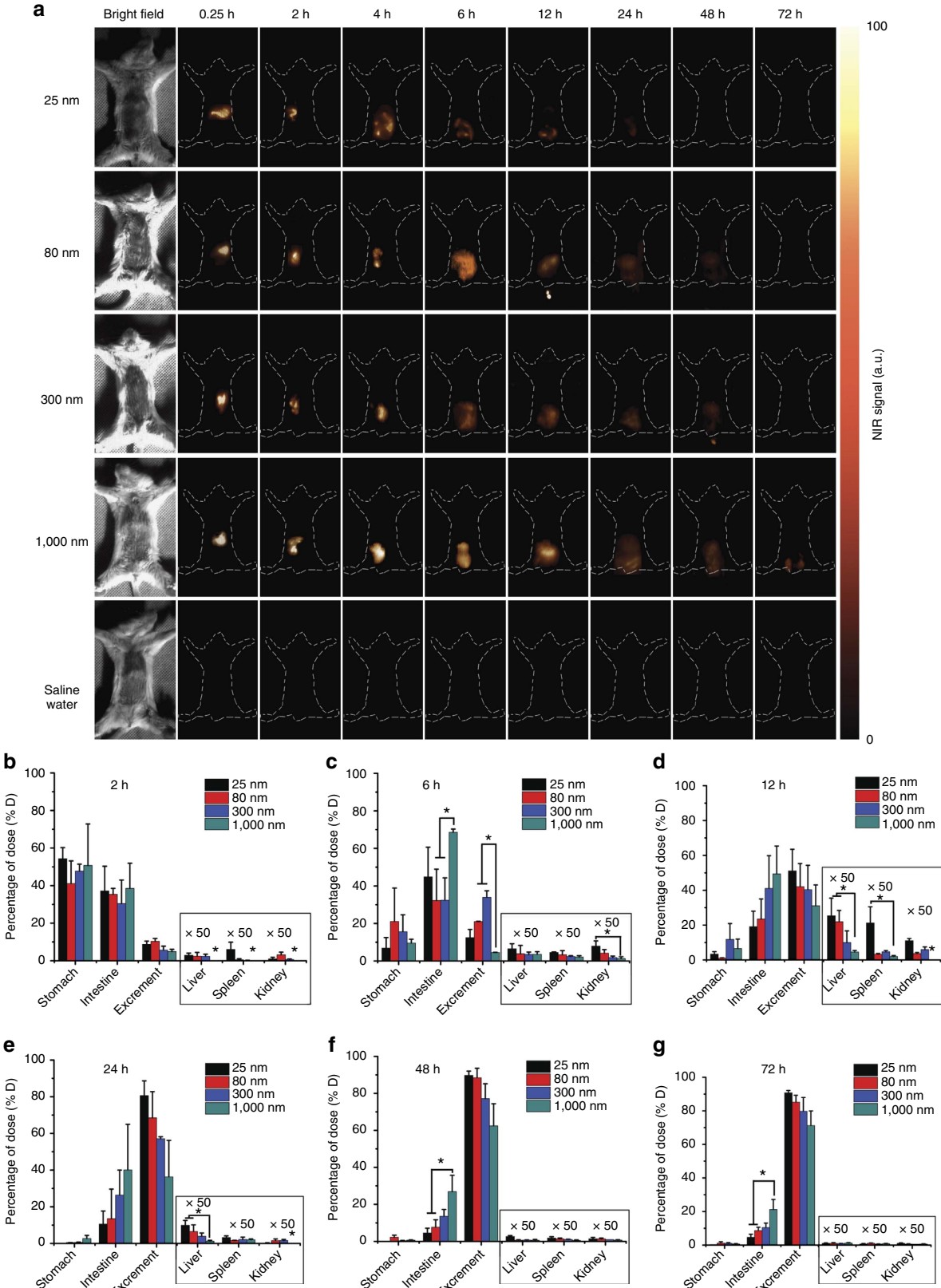

**Figure 4 | *In vivo* NIR bioimaging of the GI tract with different sized microcarrier gavaging.** (**a**) *In vivo* NIR bioimaging of mice at different times after oral gavaging with different sized particles (25, 80, 300 and 1,000 nm) and saline water as the negative control. All images were taken by an InGaAs NIR CCD camera under 808-nm laser excitation (0.2 W cm$^{-2}$). Representative images are for $n=3$ per group. (**b–g**) Biodistribution results determined by ICP measurements of the particles in various organs (stomach, intestine, liver, spleen and kidney) and excrement at different times after oral gavaging. Mean ± s.d. for $n=3$ (*$P$ value < 0.05).

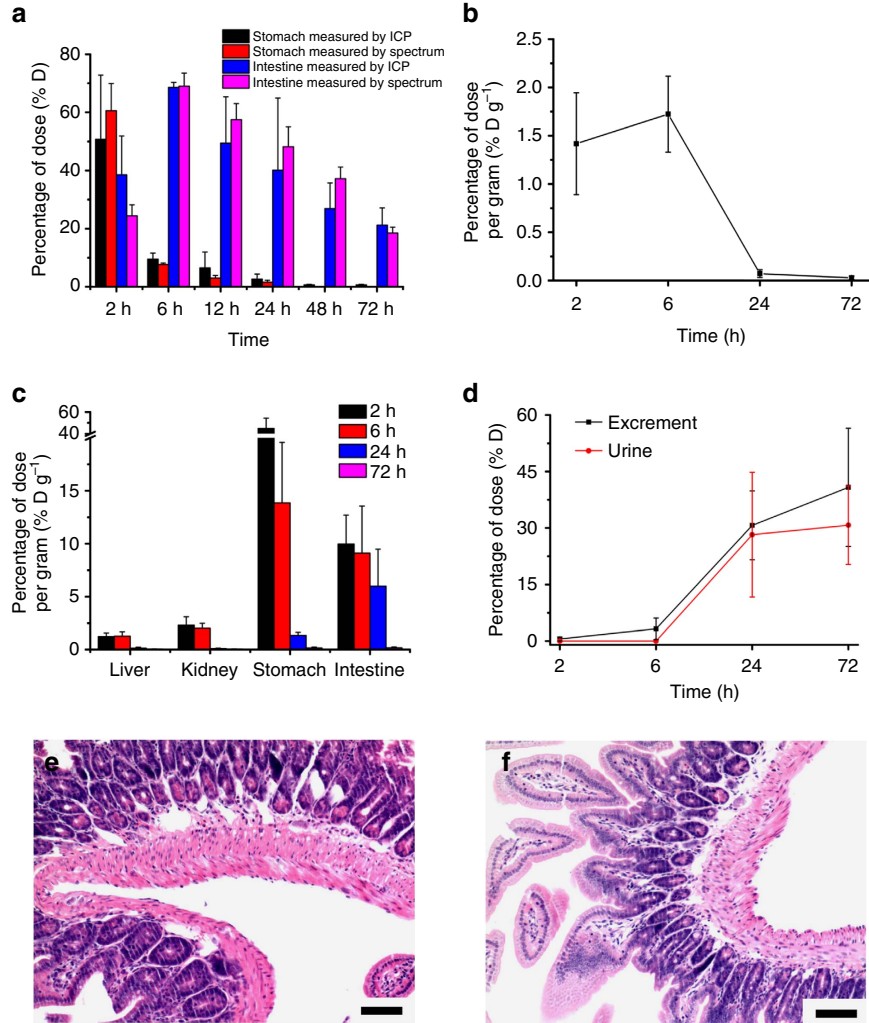

**Figure 5 | Pharmacokinetics of BSA and toxicity evaluation of the microcarriers.** (**a**) Investigation of the BSA pharmacokinetic results (stomach and intestine) by ICP and NIR imaging measurements, respectively ($P > 0.05$). Herein, 1,000-nm microcarriers were used as an example. (**b**–**d**) Time-dependent plasma concentration (**b**) tissue distribution (**c**) and cumulative excretory amounts (**d**) of [125]I-BSA in mice. (**e,f**) H&E-stained intestine sections of mice after seven days of receiving (**e**) microcarriers (50 mg kg$^{-1}$) and (**f**) saline water daily. All treatments were given via a daily oral gavage. Mean ± s.d. for $n = 3$. (**e,f**) Scale bars, 100 μm.

cumulative release amount of BSA reached $108.1 \pm 12.3$ μg at 24 h, while it merely increased to $117.3 \pm 11.3$ μg at 72 h, indicating the burst release during the first 24 h followed by a sustained release behaviour. These findings match well with the release profile of the *in vitro* experiment and prove the feasibility of our ACIE method for *in vivo* drug-release measuring in real time.

Furthermore, to investigate the activity of the released proteins from the orally delivered microcarriers, a bacterial proline-specific endopeptidase peptide (PEP) from *Myxococcus xanthus* (MX) was selected as a model enzyme. PEP is a type of enzyme that has been recently proposed as an adjuvant drug for coeliac disease therapy, but it is easily deactivated in the stomach[7,42]. PEP–NPTAT was loaded into the mesoporous microcarriers instead of BSA–NPTAT (the maximum loading capacity for PEP is 40.4 wt.%). The $K_A$ and $k_q$ values for PEP–NPTAT were measured to be $2.3 \times 10^6$ M$^{-1}$ and $2.3 \times 10^{14}$ M$^{-1}$ s$^{-1}$, respectively, which were two orders of magnitude higher than those of the BSA–NPTAT, indicating the strong affinity between PEP and NPTAT (Supplementary Fig. 14; Supplementary Table 1). To monitor the PEP activity, an FL-quenched peptide probe ((DABCYL)-LPYPQPK (Glu(EDANS)) that bore both a

fluorophore, EDANS, and the corresponding quencher DABCYL at each extremity was used as the enzymatic substrate. The FL of EDANS was largely quenched by DABCYL when the peptide structure was intact but was recovered when the probe was selectively decomposed by the released PEP. Proteolysis of this probe at pH 8 led to a 20-fold increase in the FL signal, while an unnoticeable change was observed at pH 1 and 5 (Supplementary Figs 15 and 16; Supplementary Table 4), further demonstrating that our microcarriers can provide perfect protection to the vulnerable protein drugs.

The PEP–NPTAT-loaded microcarriers were evaluated *in vivo* with an ACIE NIR-II bioimaging system using a native PEP molecule and PEP-loaded microcarriers without SSPI coatings as the control groups. The activity of the PEP was monitored by an FL-quenched probe (DABCYL)-LPYPQPK (Glu(EDANS)). The probe was gavaged 2 h before each NIR-II imaging time point (6, 24, 48 and 72 h), except for 0.25 h, when the probe was gavaged together with the microcarriers. Both the *in vivo* NIR-II bioimaging signals under 730-nm or 808-nm excitation and the FL bioimaging signals of the peptide probe under 405-nm excitation after dissection were taken at each time point (Fig. 6d–i). Similar to the results of the BSA–NPTAT-loaded

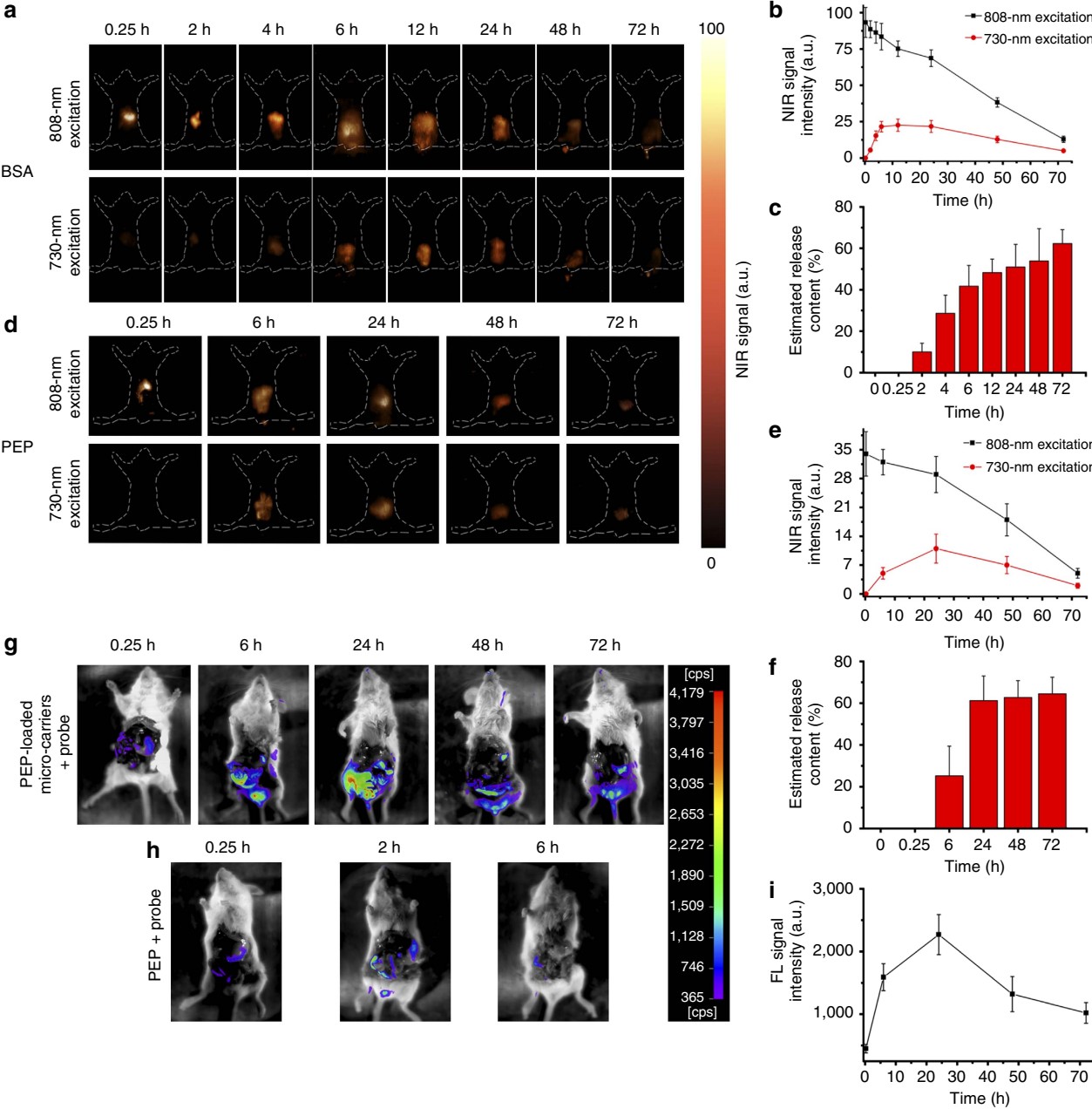

**Figure 6 | *In vivo* protein release and activity evaluation.** (**a**) NIR bioimaging of mice at different times after orally gavaging BSA–NPTAT-loaded microcarriers under 730-nm or 808-nm excitation. (**d**) NIR bioimaging of mice at different times after orally gavaging PEP–NPTAT-loaded microcarriers under 730-nm or 808-nm excitation. (**g,h**) *In vivo* bioimaging of mice after gavaging with PEP–NPTAT-loaded microcarriers and a peptide probe (**g**) and PEP molecules and a peptide probe (**h**); 405 nm was used as the excitation source. Representative images are for $n = 3$ per group. (**b,e,i**) Corresponding signal intensity curves of (**a,d,g**) respectively. (**c–f**) *In vivo* release percentages ($Q_{(\tau)}$) of BSA–NPTAT (**c**) and PEP–NPTAT (**f**) from microcarriers calculated using equation (1). Mean ± s.d. for $n = 3$.

microcarriers, the NIR-II bioimaging signals under 730-nm and 808-nm excitation differed significantly during the first 6 h then became closer gradually. Therefore, the PEP release amount could be predicted using the same equations (equations (1) and (2)) for the BSA–NPTAT system. As shown in Fig. 6f and Supplementary Table 5, the calculated $Q$ and $M_t$ values were 25% and 54.8 ± 10.2 μg, respectively, at 6 h. Then, a burst release was observed during the following 18 h, and the values reached ∼60% and 125.7 ± 11.8 μg at 24 h, respectively. According to previous reports, only 1 μg of PEP in the GI tract should be enough to cause therapeutic effects for a diseased mouse[42]. Therefore, the drug delivery capacity of PEP from the microcarriers was

sufficiently high for effective treatment. Furthermore, FL recovery of the peptide was utilized to assess the activity of the released PEP (Fig. 6g–i; Supplementary Fig. 17). In comparison, the control groups including native PEP and the microcarriers without SSPI coatings showed little activity because of the PEP degradation in the GI tract. While for the PEP–NPTAT-loaded microcarriers that had the same PEP amount as the native PEP, the FL enhancement of the peptide reached ∼8-fold at 24 h, and a 3-fold enhancement was even observed at 48 h, suggesting that the sustained releasing behaviour and retained activity of the PEP enzyme can be realized by the microcarrier drug delivery system.

**Table 2 | Cumulative release amount of BSA calculated by the equation (2).**

| Time (h) | Cumulative release amount (µg) |
|---|---|
| 0.25 | $0 \pm 0$ |
| 2 | $22.5 \pm 5.6$ |
| 4 | $62.8 \pm 11.9$ |
| 6 | $90.3 \pm 11.5$ |
| 12 | $103.3 \pm 5.7$ |
| 24 | $108.1 \pm 12.3$ |
| 48 | $111.9 \pm 14.4$ |
| 72 | $117.3 \pm 11.3$ |

## Discussion

The microcarriers reported here provide many attractive features for oral drug delivery, such as high loading capacities, effective protection and high therapeutic indexes. Moreover, the properties of the microcarriers, such as a tunable pore size (2.5–12 nm, Supplementary Figs 18 and 19), aspect ratio (2:1–10:1, Supplementary Fig. 20), surface charge (negative or positive) and surface hydrophobicity, endow them with great potential in the field of drug delivery. To load a desired protein drug, there are several criteria for the design of a microcarrier: the pore size should be larger than the desired protein drug, the aspect ratio should be adequate to entrap the whole protein molecule, and the microcarrier and desired protein drug should have opposite surface charges during loading.

The majority of the available evidence in the literature suggests that the absorption of particles predominantly occurs in the intestinal lymphatic tissues (that is, Peyer's patches)[43,44]. The epithelial cell layer overlying the Peyer's patches contains specialized M cells that are believed to be transcytotic[45]. Aside from particle size, the particle absorption rate as well as the absorption efficiency of the Peyer's patches have also been shown to be affected by the surface properties of the particles, such as their surface charge and hydrophobicity[46]. Generally, hydrophobic particles have been absorbed more readily than hydrophilic ones[47], while positively charged particles have increased the mucoadhesive properties of the particles[7]. In this study, we found that the retention time and deposition amount in the visceral organs was size dependent. Notably, the smaller microcarriers, such as the 25-nm microcarriers, showed a relatively short retention times (<24 h) and high uptake amounts (∼1.15% at 12 h, Fig. 4b–g). Therefore, further toxicity investigations of the small microcarriers are required before they can be utilized because these microcarriers in the bloodstream may induce haemolysis, allergic reactions and even some severe diseases. In contrast, the 1,000-nm microcarriers used in our protein delivery experiments exhibited many advantages as good oral drug carriers, such as extended retention times reaching 72 h and minimal deposition (<0.1%) in the visceral organs. The long retention behaviour of the drug carrier can be exploited for prolonged drug delivery systems, which have more flexibility in their dosage design than conventional drug delivery systems[48]. In addition, we also noticed that the mice in our experiments did not show any signs of discomfort and that their weight gain was normal (average weight increase of ∼200% after three months). Meanwhile, after 7 days of receiving microcarriers daily, the histopathological examination showed that the morphologies of the intestinal tissues and tight junctions were intact without any signs of degeneration (Fig. 5e,f), further indicating the low health risk caused by the microcarriers.

In conclusion, the ACIE method reported here is an NIR bioimaging technique that is based on the multiexcitation

properties of the NIR-II DCNPs (Supplementary Fig. 21). Compared with the traditional radioisotopic method, the ACIE NIR bioimaging technique is non-invasive for *in vivo* detection. However, only *in situ* drug-release information can be reflected by the ACIE system. Other techniques, such as radioisotopic methods, are still required to track the traces and activity of the released drugs. To sum up, we expect that the ACIE technique will be strongly complementary to the traditional radioisotopic tracing method to realize non-invasive *in vivo* semi-quantitative drug-release monitoring. In addition, we believe that this ACIE technique not only holds great potential for monitoring the oral drug delivery process but also propels the development of the *in situ* NIR sensing of biomarkers such as DNA, redox species or specific enzymes. In principle, the ACIE technique will expand the library of NIR sensors because many commercial visible and NIR-I reporters can also be used for NIR-II sensing if they share a similar absorption wavelength region with the DCNPs.

## Methods

**Preparation of microcarriers and protein drug loading.** The material synthesis, surface modification and characterization of $SiO_2$-Nd@$SiO_2$@m$SiO_2$-$NH_2$@SSPI core/shell structured microcarriers are described in detail in the Supplementary Methods.

**Investigation of the interactions between proteins and NPTAT.** Different amounts of NPTAT (20 nM in aqueous solution) were added dropwise to a solution of BSA (150 nM) in Tris–HCl buffer at various BSA/NPTAT molar ratios (14.7, 7.4, 4.9, 3.7 and 2.9). The FL spectra of these samples were recorded under 280-nm excitation (Supplementary Fig. 7a). As shown in Supplementary Fig. 7b, the quenching data follow the Stern–Volmer equation (equation (3)) and its double-reciprocal form, the Lineweaver–Burk plot (equation (4)), where $F_0$ and $F$ are the intensities of FL in the absence and presence of the NPTAT, respectively. [Q] is the concentration of the NPTAT, and $K_A$ is the binding constant that reflects the degree of interaction between the NPTAT and BSA. $\tau_s$ is the lifetime of the BSA in the absence of the quencher (equation (5)).

$$F_0/F = 1 + K_A[Q] \tag{3}$$

$$(F_0 - F)^{-1} = F_0^{-1} + K_A^{-1}F_0^{-1}[Q]^{-1} \tag{4}$$

$$k_q = K_A/\tau_s \tag{5}$$

The $K_A$ value was calculated to be $7.20 \times 10^4 \, M^{-1}$ according to the slope in Supplementary Fig. 7b. Since $\tau_s$ was on the order of $10^{-8}$ s, $k_q$ was calculated to be $7.2 \times 10^{12} \, M^{-1} \, s^{-1}$ according to equation (4), which is much higher than the normal value for dynamic quenching (*ca.* $10^{10} \, M^{-1} \, s^{-1}$), indicating strong interactions between BSA and NPTAT.

The interaction between the PEP and NPTAT was investigated using the same method.

***In vitro* drug-release measurements.** *In vitro* release experiments were performed in simulated gastric fluid (pH 1 in 0.1 M HCl), duodenum fluid (pH 5 in PBS buffer) and intestinal fluid (pH 8 in PBS buffer) to mimic protein release in the stomach, duodenum and intestine ($n = 3$), respectively. The detailed compositional information of the simulated body fluids is described in the Supplementary Methods. First, 20 mg of protein-loaded $SiO_2$-Nd@$SiO_2$@m$SiO_2$-$NH_2$@SSPI NPs was added to 10 ml of simulated gastric fluid at 37 °C with slight shaking. After 2 h, the suspension was centrifuged, and the sediments were redispersed in 10 ml of pH 5 buffer to simulate the duodenum. After another 2 h, the sediments were transferred into simulated intestinal fluid (pH 8) to monitor the drug release for 16 h. At selected time intervals, 0.25 ml of the samples was withdrawn and immediately replaced with an equal volume of the simulated fluid. The samples were centrifuged at 3,000 r.p.m., the supernatants were analysed on a ultraviolet–vis spectrophotometer, and the sediments were redispersed in MES buffer (pH 5) for the FL measurements.

**Biodistribution investigation by NIR bioimaging and ICP.** All the following animal procedures were in agreement with the guidelines of the Institutional Animal Care and Use Committee of Fudan University and performed in accordance with the institutional guidelines for animal handling. All of the animal experiments were permitted by the Shanghai Science and Technology Committee.

*In vivo* NIR bioimaging was performed with a modified NIRvana CCD camera (Princeton Instruments) that had an external 0–2 W adjustable 808-nm CW laser as the excitation source, in combination with a longpass optical filter (1000 LP from Chroma). A quantity of 1 mg of 25, 80, 300 and 1,000 nm SSPI-modified microcarriers in 0.2 ml of saline water (pH adjusted to 4) was imbued into the

stomachs of mice (5–6 weeks, female, $n = 3$ per group) using a gastric syringe. A quantity of 0.2 ml of saline water without particles was used as the control group. Then, the optical whole-body bioimaging results of the mice were recorded at each time point by the modified NIRvana CCD camera.

Kunming mice (5–6 weeks, female) were fasted for 12 h before the experiments. Then, the animals were randomized into five groups: 25, 80, 300, 1,000 nm and control (saline water) ($n = 3$ per group). The mice were killed at 2, 6, 12, 24, 48 and 72 h after oral administration of the particles at a dosage of 50 mg kg$^{-1}$. The excrement and main organs (stomach, intestine, liver, spleen and kidney) of each group were collected. In addition, the concentrations of Si in each tissue were measured by ICP.

**In vivo GI drug-release monitoring through NIR-II bioimaging.** The 1,000-nm microcarriers were used in the following experiment.

The procedure for the *in vivo* release prediction of BSA–NPTAT or PEP–NPTAT is similar to the biodistribution investigation procedure for NIR-II bioimaging, except that bioimaging was also taken at each time point by simply switching the laser source under 730-nm excitation ($n = 3$ per group). Because the NIR-II signals excited at 808 nm remained constant as the NPTAT loading amount increased, they reflected the real quantity of the microcarriers as reference signals. The calculation method is shown in equation (1).

**In vivo GI enzyme activity monitoring by NIR-II bioimaging.** The *in vivo* activity of PEP was monitored by the FL-quenched peptide (DABCYL)–LPYPQPK (Glu(EDANS)). The procedure is also similar to the biodistribution investigation by NIR *in vivo* bioimaging, except that 0.1 mg of (DABCYL)–LPYPQPK (Glu(EDANS)) was imbued 2 h before imaging. In the 0.25-h group, the peptide was gavaged together with the particles. Notably, the pure PEP molecules and microcarriers without an SSPI coating served as the control groups ($n = 3$ per group). The optical whole-body bioimaging of mice was recorded on an LB 983 NightOWL II instrument equipped with 405-nm excitation and 525-nm emission filters.

**Data availability.** The data that support the findings of this study are available from the corresponding author upon reasonable request.

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

## Acknowledgements

The work was supported by the China National Key Basic Research Program (973 Project) (No. 2013CB934100), NSFC (Grant Nos 21322508 and 21210004), Shanghai Shuguang Program and China Postdoctoral Science Foundation (2015M570327). We thank Professor Jianhua Zhu from the School of pharmacy, Fudan University for the pharmacokinetic experiments using a radioisotopic tracing method.

## Author contributions

F.Z. and R.W. conceived the project and designed the experiments. R.W., L.Z., W.W., X.L. and F.Z. were primarily responsible for the data collection and analysis. F.Z. and R.W. prepared the figures and wrote the main manuscript text. All authors contributed to the discussions and manuscript preparation.

## Additional information

**Competing financial interests:** The authors declare no competing financial interests.

