## [Peer Review File · Nature Communications]

Reviewers' comments:

Reviewer #1 (Remarks to the Author):

The manuscript titled 'In-vivo gastrointestinal drug release monitoring through second near-infrared window fluorescent bioimaging with orally delivered micro-carrier' reports a novel microcarrier system of lanthanide-based downconversion nanoparticles and silica particles for bioimaging of gastrointestinal drug release using second near infrared window fluorescent signal. Wang et al., demonstrate that these microcarrier system show a 72 h residence time in the gastrointestinal tract and can be used for bioimaging by exploiting the NIR-II signals from two different excitation sources.

Overall this is an interesting study that evaluates image-guided oral delivery using microparticle systems. Several major issues need to be addressed prior to publication.

Comments:

1. The authors indicate that the microcarrier system has no toxic effects based on the normal weight gain and no signs of discomfort. However, further studies such as the effect on intestinal tissue morphology and tight junctions would be required as a proof of concept to indicating no toxic effects of this microcarrier system on the gastrointestinal tract.
2. Quantitative biodistribution analysis should be carried out to correlate with qualitative imaging studies.
3. Authors need to mention the sample size (n) number in the experimental methodology section wherever applicable.
3. With respect to characterization of this microcarrier system, it is essential to determine and discuss the maximum loading ability of this microcarrier system with respect to a model proteins used in the study.
4. The y-axis for Figure 5 b, e and h needs to be well defined.
5. The amount of protein/drug delivered at the desired location is an important factor in evaluating the efficacy of the formulation as a carrier system. The authors demonstrate that the microcarrier system show 65% release of its contents in the intestinal sections at pH 8. Furthermore, the authors have confirmed the protein encapsulated is protected from degradation and retains its activity when released from the carrier system at pH 8. It could be ideal to demonstrate that the amount of (model) protein released from the microcarrier system in the intestinal section at pH 8 is suitable to bring about therapeutic effect.
6. Language errors need to be corrected.

Reviewer #2 (Remarks to the Author):

In this paper, Wang and co-workers fabricated a novel kind of DCNPs loaded mesoporous microparticles as an efficient drug delivery carrier for NIR-II monitored oral drug delivery. Taking advantage of the multi-absorption property of lanthanide ions, a novel absorption competition system has been utilized to predict the delivered content in real time. To my point of view, this system can be seen as a big breakthrough in this optical biosensing research area. It provides a feasible way for monitoring or detecting biomolecule in NIR-II region with deeper penetration depth and higher

resolution, compared with those fluorescent probes working at traditional visible or NIR-I region. I therefore strongly recommend its publication in Nature communications. Some minor revisions that could further improve the technical content of this work are outlined below.

Comments:

1. The release content monitoring technique used in this work actually is semi-quantitative, because the released content and the change of NIR-II signals are linked by the amount of NPTAT, rather than directly feedback. Therefore, I suggest that it will be proper to replace the phrase "quantitative" with "semi-quantitative" in the whole manuscript.
2. The absorption spectrum of DCNPs in Figure 3a is unclear, please provide a clearer image to displace it.
3. In Figure 4c, there are some mistakes displayed in the x-axis. "Excrement" is partially shadowed.

Technical comments:

1. The emission of Nd doped DCNPs consists of several bands. Please provide the proportion of 1060 nm emission intensity among these bands at different power density.
2. Neodymium ion possesses multi-absorption property. Is there also multi-excitation property?
3. Whether the NIR cyanine dyes can be used in absorption competition system or not? They have wide varieties and are commercial available with controlled absorption wavelength.
4. Why is the long retention time in GI tract helpful in oral drug delivery? Please explain it.
5. Why these particles cannot be absorbed by the body? Besides the particle size, I think they might be affected by the surface properties.

Reviewer #3 (Remarks to the Author):

This study reports a new micro-carrier to facilitate oral protein drug delivery. This micro-carrier protects drugs during passage through the stomach and allows for tracking of micro-carrier anatomical location via near-IR fluorescence. The particles are made and thoroughly characterized using standard techniques. An NPTAT dye was co-loaded with BSA protein and used as a proxy to monitor protein PK and in vitro release profiles monitored at several pH. Particles of varying size were gavaged into mice and rats and found to clear >80% particles in 24 hours. Finally, a model enzyme PEP was administered with NPAT and a peptide substrate appending with a pair of FRET probes to discriminate between cleaved and uncleaved peptide.

This is an important area of research and a lot of work was performed in this study. The authors present an interesting approach for both micro-carrier design and PK monitoring of the micro-carriers, using lanthanide-based down-conversion nanoparticles with near-IR fluorescence and a mesoporous structure. Non-invasive monitoring via fluorescence is very important and the key advance here is the enhanced bio-imaging, including 1-2 cm tissue penetration and sub-10 um resolution combined with protein drug delivery. However several key questions related to feasibility of their approach for drug delivery were not addressed. In particular, their protein drug PK data is not clear (where did the protein go, how fast and while retaining how much activity) and importantly, was not validated using standard techniques.

Major comments:

1. While the PK of the microcarriers is an important questions, the PK/ PD of the protein drug is at least, if not more important. Despite the title, the protein PK has not really been monitored or reported. Why was the reporting fluorophore not covalently coupled to BSA to be a more robust report of protein location? The in vitro BSA and NPTAT release rates should be quantified and compared in the text as part of the validation (data in Figure 3d; lines 190-200). Since the PEP protein is a

different protein with different surface hydrophobicity/ charge characteristics, the in vitro loading and release profiles also need to be measured and reported for this protein. Figure 5, which ostensibly shows the in vivo protein PK data are unclear and unconvincing. How much protein is released, where does it go (is it also restricted to the intestinal lumen? If not, why not?), does it retain biological activity or is it fragmented? These PK/ PD data need to be validated with conventional methods, such as ELISA or ¹²⁵I and activity assays.

2. Please clarify the goals of this system: is the intent of this system to deliver drugs to the bloodstream or just to the intestine? What are the potential mechanisms by which the particles could transfer from the intestine into the blood or otherwise access the organs? What are the likely transport rates via these mechanisms?

3. There is very little discussion of the data, short-comings or comparisons with current or competing methods in development. In addition to these items, please discuss possible sources of toxicity of these microcarriers, transfer from the intestine to the blood, effects of microparticle size and considerations for using different proteins, etc.

Detailed comments:

4. According to Fig. 4a, at 0.25h, the NIR signal in 1000 nm particles group is higher than the other groups, especially 25nm group. It seems like the original amounts of DCNPs in each groups are different. Thus, the larger the particles, the more the DCNPs will be in the particles. The conclusion that larger particles can stay longer in GI tract than smaller particles could not be made without standardizing NIR signal. It will be better to standardize the NIR signal in each groups with relative DCNPs amount.

5. In Fig. 5g, it will be better to choose PEP loaded micro-carriers without SSPI covered + probe to be control. This could prove it is because SSPI that retain PEP in micro-carriers until releasing in intestine. This control could prove the higher FL intensity in PEP loaded micro-carriers+ probe group is not because micro-carriers itself.

6. What is the size of the pores in the microparticles that allow it to load protein drug? Can these sizes be tuned or will pore size limit the choice of protein drug to load? What are the expected effects of aspect ratio, surface charge and surface hydrophobicity? (for instance, an antibody has a hydrodynamic radius of ~ 10 nm). K_a and k_q and equations (1-4) should be explained more clearly.

7. The use of mice versus rats should be clarified in the results (lines 207 and 228, 358)

Point-by-Point Response to Reviews

Reviewers' comments:

Reviewer #1 (Remarks to the Author):

The manuscript titled 'In-vivo gastrointestinal drug release monitoring through second near-infrared window fluorescent bioimaging with orally delivered micro-carrier' reports a novel microcarrier system of lanthanide-based downconversion nanoparticles and silica particles for bioimaging of gastrointestinal drug release using second near infrared window fluorescent signal. Wang et al., demonstrate that these microcarrier system show a 72 h residence time in the gastrointestinal tract and can be used for bioimaging by exploiting the NIR-II signals from two different excitation sources.

Overall this is an interesting study that evaluates image-guided oral delivery using microparticle systems. Several major issues need to be addressed prior to publication.

Comments:

1. The authors indicate that the microcarrier system has no toxic effects based on the normal weight gain and no signs of discomfort. However, further studies such as the effect on intestinal tissue morphology and tight junctions would be required as a proof of concept to indicating no toxic effects of this microcarrier system on the gastrointestinal tract.

Response: Thanks so much for the useful comments and suggestions. In the revised manuscript, histopathological examination of the intestine was carried out to check the tissue morphology and intestinal tight junctions after oral administration of micro-carriers. All of them were found to be intact without any signs of degeneration, indicating that micro-carriers did not induce damage in these tissues.

We have added the following contents in the revised manuscript on line 351, page 12:

“Meanwhile, after seven days of daily receiving micro-carriers, histopathological examination showed that the morphology of intestinal tissue and tight junctions were intact without any signs of degeneration (Fig. 5e-f), further indicating little healthy risky was caused by the micro-carriers.”

And we have added Fig. 5e-f in the revised manuscript with the following figure caption “H&E-stained intestine sections of mice after seven days of daily receiving (e) micro-carriers (50 mg kg⁻¹) and (f) saline water. All treatments were given via a daily oral gavage.”

2. Quantitative biodistribution analysis should be carried out to correlate with qualitative imaging studies.

Response: Thanks for the comments. We have added **Fig. 5a** to illustrate the correlation of the biodistribution analysis between inductively coupled plasma atomic emission spectrometry (ICP) and NIR imaging measurement.

We have added the following contents in the revised manuscript on line 231, page 8:
“By comparing the biodistribution (% ID) results measured by ICP and NIR bioimaging techniques, we found that the differences between them were not significant ($p > 0.05$) (Fig. 5a), demonstrating the NIR-II bioimaging approach is rather convincing for *in vivo* monitoring the biodistribution of the micro-carriers.”.

3. Authors need to mention the sample size (n) number in the experimental methodology section wherever applicable.

Response: We appreciate the reviewer for the useful suggestion. In the revised manuscript and supplementary information, we have added the detailed sample size information in the methodology section (revised manuscript on line 426, page 15 and supplementary information on line 12, page 27) as following: “1000 nm micro-carriers were used in the following experiment.”
We also added similar description in the figure caption of Fig. 5a: “Herein, 1000 nm micro-carriers were used as an example.”

4. With respect to characterization of this microcarrier system, it is essential to determine and discuss the maximum loading ability of this microcarrier system with respect to a model proteins used in the study.

Response: Thanks so much for the comment. We have added the maximum loading capacity in the revised manuscript as following: “maximum loading capacity for BSA is 47.4 wt. %” (on line 186, page 7) and “maximum loading capacity for PEP is 40.4 wt. %” (on line 306, page 11).

5. The y-axis for Figure 5 b, e and h needs to be well defined.

Response: Thanks for the comments. Fig. 5 has been renamed as Fig. 6 in the revised manuscript. The signal intensities in Figure 6b,e and h have already been normalized. In the revised manuscript, we have renamed y-axis of Figure 6b,e with “NIR signal intensity (a.u.)”, while that of Figure 6h with “FL signal intensity (a.u.)”. And we also provided the normalized intensity number in y-axis of these revised figures.

5. The amount of protein/drug delivered at the desired location is an important factor in evaluating the efficacy of the formulation as a carrier system. The authors demonstrate that the microcarrier system show 65% release of its contents in the intestinal sections at pH 8. Furthermore, the authors have confirmed the protein encapsulated is protected from degradation and retains its activity when released from the carrier system at pH 8. It could be ideal to demonstrate that the amount of (model) protein released from the microcarrier system in the intestinal section at pH 8 is suitable to bring about therapeutic effect.

Response: We appreciate the reviewer for the useful suggestion. To further demonstrate the exact amount of BSA released from the micro-carriers, we have added the following contents in the revised manuscript on line 279, page 10:

“Based on the obtained release percentage $Q_{(t)}$, cumulative release amount of BSA in real time

can be derived by the following equation:

$$M_{(\tau)} = M_{(\tau-1)} + \frac{F_{808,(\tau)}}{F_{808,0}} m\theta(Q_{(\tau)} - Q_{(\tau-1)}) \quad (2).$$

Where $M_{(\tau)}$ and $M_{(\tau-1)}$ were the cumulative release amount of BSA at two closed time point (τ and $\tau-1$). $Q_{(\tau)} - Q_{(\tau-1)}$ reflected the release percentage during this time period (from time ($\tau-1$) to time (τ)). $F_{808,(\tau)}$ and $F_{808,0}$ are used to represent the intensity of NIR-II signals under 808 nm excitation for drug loaded micro-carriers at time (τ) and initial time point just after gavaging. ($F_{808,(\tau)}$ and $F_{808,0}$ can be obtained from Fig. 6b). Since the in-vivo delivery of micro-carriers is a dynamic process, the micro-carriers will be cleaned out gradually, the real-time micro-carrier percentage in the body can be calculated from $\frac{F_{808,(\tau)}}{F_{808,0}}$. Coefficient m and θ are the initial micro-carriers mass and drug loading percentage. Therefore, the release amount of BSA from micro-carriers during a specific period (from time ($\tau-1$) to time (τ)) can be calculated from $\frac{F_{808,(\tau)}}{F_{808,0}} m\theta(Q_{(\tau)} - Q_{(\tau-1)})$. Taking the cumulative drug release calculation from 6 h to 12 h as an example, $Q_{(12\text{ h})}$ and $Q_{(6\text{ h})}$ value were calculated to be 48% and 42% according to Eq. (1), respectively (Fig. 6c). Meanwhile, $F_{808,(12\text{ h})}$ and $F_{808,0}$ value were measured to be 75.2 and 93.3 (derived from Fig. 6b), so the $M_{(12\text{ h})}$ can be estimated to be $100.0 \pm 5.7 \mu\text{g}$. The calculated cumulative drug release amount with microcarriers were showed in Table 2. It was shown that the cumulative release amount of BSA can reach to $105.6 \pm 12.3 \mu\text{g}$ at 24 h, while it merely increased to $111.1 \pm 11.3 \mu\text{g}$ at 72 h, indicating the burst release at first 24 h, followed by a sustained release behavior. These findings matched well with the release profile in *in vitro* experiment, and also proved the feasibility of our ACIE method for the *in vivo* drug release measuring in real time.”

The protein drugs vary widely in the therapeutic effective dose. As for PEP introduced in the present work, only 25 mU ml^{-1} (equals to $\sim 1 \mu\text{g ml}^{-1}$) of PEP in the GI tract would be enough to bring about therapeutic effect for celiac disease according to previous reports (Fuhrmann, G., et al. Nat. Chem. **5**, 582-589 (2013)). Since the cumulative PEP release amount can reach to $54.8 \pm 10.2 \mu\text{g}$ with the micro-carrier drug delivery system, the drug delivery capacity for PEP were considerably suitable to bring about effective treatment. In order to make this point clearer for readers, we added the following description in the revised manuscript on line 330, page 11. We also added Supplementary Table 4 in the revised supplementary information to show the cumulative PEP release amount at different time point.

“According to previous reports, only 25 mU ml^{-1} (equals to $\sim 1 \mu\text{g ml}^{-1}$) of PEP in the GI tract would be enough to bring about therapeutic effect for celiac disease. Therefore, the drug delivery capacity our micro-carriers for PEP were considerably suitable to bring about effective treatment.”

6. Language errors need to be corrected.

Response: Thanks so much for the reviewer’s comments. We have polished our manuscript carefully and corrected the grammatical, styling, and mistyped errors found in our manuscript. Detailed modifications are also provided as following.

We corrected the following mistake on page 5, line 151: “acceptor” was changed to “acceptors”.

We corrected the following mistake on page 6, line 161: “could be” was changed to “was”

We corrected the following mistake on page 7, line 207: “extending” was changed to “extended”.

We corrected the following mistake on page 7, line 209: “is also correlated well the spectroscopy results” was changed to “also correlated well with the spectroscopy results”.

We corrected the following mistake on page 6, line 154: “not any” was changed to “none”.

We corrected the following mistake on page 6, line 169: “unconspicuous” was changed to “inconspicuous”.

We corrected the following mistake on page 8, line 243: “microcarrier” was changed to “micro-carrier”.

We corrected the following mistake on page 10, line 307: “are” was changed to “were”.

We corrected the following mistake on page 11, line 303 and page 11, line 305: “is” was changed to “was”.

Reviewer #2 (Remarks to the Author):

In this paper, Wang and co-workers fabricated a novel kind of DCNPs loaded mesoporous microparticles as an efficient drug delivery carrier for NIR-II monitored oral drug delivery. Taking advantage of the multi-absorption property of lanthanide ions, a novel absorption competition system has been utilized to predict the delivered content in real time. To my point of view, this system can be seen as a big breakthrough in this optical biosensing research area. It provides a feasible way for monitoring or detecting biomolecule in NIR-II region with deeper penetration depth and higher resolution, compared with those fluorescent probes working at traditional visible or NIR-I region. I therefore strongly recommend its publication in Nature communications. Some minor revisions that could further improve the technical content of this work are outlined below.

Comments:

1. *The release content monitoring technique used in this work actually is semi-quantitative, because the released content and the change of NIR-II signals are linked by the amount of NPTAT, rather than directly feedback. Therefore, I suggest that it will be proper to replace the phrase “quantitative” with “semi-quantitative” in the whole manuscript.*

Response: We accept this comment. The “quantitative” has been changed to “semi-quantitative” in the whole manuscript.

2. *The absorption spectrum of DCNPs in Figure 3a is unclear, please provide a clearer image to displace it.*

Response: Thanks for the comment. In order to make the spectrum more identifiable, we have provided the **partial enlarged figure** of absorption spectrum in the revised manuscript as **Supplementary Fig. 5**.

We have added Supplementary Fig. 5 with the following figure caption:
“Supplementary Fig. 5. Partial enlarged figure of Figure 3a.”

3. *In Figure 4c, there are some mistakes displayed in the x-axis. “Excrement” is partially*

shadowed.

Response: We appreciate this suggestion. We have modified this figure in the revised manuscript (Figure 4c).

Technical comments:

1. The emission of Nd doped DCNPs consists of several bands. Please provide the proportion of 1060 nm emission intensity among these bands at different power density.

Response: We appreciate this suggestion. In the revised manuscript, we summarized the proportion of three primary emission bands (860, 1060 and 1340 nm) of Nd doped DCNPs excited by 730 or 808 nm at different power density.

We have added Supplementary Fig. 11 and Supplementary Table 2 with the following figure caption: “Supplementary Fig. 11. A typical downconversion emission spectrum of NaGdF₄:5%Nd@NaGdF₄ DCNPs excited by 730 nm or 808 nm.”

“Supplementary Table 2. The proportion of primary emission bands (860, 1060 and 1340 nm) of NaGdF₄:5%Nd@NaGdF₄ DCNPs excited by 730 or 808 nm at different power density.”

2. Neodymium ion possesses multi-absorption property. Is there also multi-excitation property?

Response: Thanks for the comment. Actually, we found that there are four primary excitation bands (730, 800, 862 and 894 nm) in NIR region. All of these excitation bands can lead to 1060 nm emission. In order to make this point clearer for readers, we have added Supplementary Fig. 19 in the revised supplementary information with the following figure caption:

“Supplementary Fig. 19. Multi-excitation property of the NaGdF₄:5%Nd@NaGdF₄ DCNPs. The NIR emission spectrum of NaGdF₄:5%Nd@NaGdF₄ DCNPs (20 mg ml⁻¹, dispersed in cyclohexane) excited by Xenon lamp at different wavelength.”

3. Whether the NIR cyanine dyes can be used in absorption competition system or not? They have wide varieties and are commercial available with controlled absorption wavelength.

Response: Thanks so much for the comment. As discussed in the manuscript, there are some requirements for the absorbing competition acceptor, such as absorption overlapping with DCNPs, little fluorescence and strong affinity with drugs. Obviously, the biggest challenge for commercial NIR cyanine dyes (like IR-780 or IR-820) is their strong NIR emission, which would significantly decrease the signal/noise ratio of the DCNPs, as they always share the similar excitation bands with lanthanide doped DCNPs

4. Why is the long retention time in GI tract helpful in oral drug delivery? Please explain it.

Response: The extended retention time in GI tract is supposed to be a solution to “medication non-adherence”, which is a major barrier to effective clinical care. In developed nations, adherence to long-term therapies is only 50%, and it is much lower in developing countries and in people who take multiple drugs with complex dose regimens (Traverso, G. & Langer, R. Special delivery for

the gut. *Nature* **519**, S19-S19 (2015).). We added the following description in the revised manuscript on line 355, page 12:

“The extended retention time in GI tract with little toxic property is supposed to be a solution to “medication non-adherence”, which is a major barrier to effective clinical care.”

5. Why these particles cannot be absorbed by the body? Besides the particle size, I think they might be affected by the surface properties.

Response: The absorption principle of particles by the GI tract is very complicated. Generally, the extent of particle uptake in GI tract is size dependent. A lot of previous works showed that the size threshold for GI uptaking is 100-1000 nm (Hongming Chen and Robert Langer, *Advanced Drug Delivery Reviews* 34 (1998) 339–350). Aside from particle size, **the surface properties** of the particles also affect GI uptake, such as surface charge and hydrophobicity. However, it is difficult to determine which factor dominate the absorption property of our micro-carriers, as the surface characteristics changes a lot during the oral drug delivering process.

Reviewer #3 (Remarks to the Author):

This study reports a new micro-carrier to facilitate oral protein drug delivery. This micro-carrier protects drugs during passage through the stomach and allows for tracking of micro-carrier anatomical location via near-IR fluorescence. The particles are made and thoroughly characterized using standard techniques. An NPTAT dye was co-loaded with BSA protein and used as a proxy to monitor protein PK and in vitro release profiles monitored at several pH. Particles of varying size were gavaged into mice and rats and found to clear >80% particles in 24 hours. Finally, a model enzyme PEP was administered with NPAT and a peptide substrate appending with a pair of FRET probes to discriminate between cleaved and uncleaved peptide.

This is an important area of research and a lot of work was performed in this study. The authors present an interesting approach for both micro-carrier design and PK monitoring of the micro-carriers, using lanthanide-based down-conversion nanoparticles with near-IR fluorescence and a mesoporous structure. Non-invasive monitoring via fluorescence is very important and the key advance here is the enhanced bio-imaging, including 1-2 cm tissue penetration and sub-10 um resolution combined with protein drug delivery. However several key questions related to feasibility of their approach for drug delivery were not addressed. In particular, their protein drug PK data is not clear (where did the protein go, how fast and while retaining how much activity) and importantly, was not validated using standard techniques.

Major comments:

1. While the PK of the microcarriers is an important questions, the PK/ PD of the protein drug is at least, if not more important. Despite the title, the protein PK has not really been monitored or reported. Why was the reporting fluorophore not covalently coupled to BSA to be a more robust report of protein location? The in vitro BSA and NPTAT release rates should be quantified and compared in the text as part of the validation (data in Figure 3d; lines 190-200). Since the PEP

protein is a different protein with different surface hydrophobicity/ charge characteristics, the in vitro loading and release profiles also need to be measured and reported for this protein. Figure 5, which ostensibly shows the in vivo protein PK data are unclear and unconvincing. How much protein is released, where does it go (is it also restricted to the intestinal lumen? If not, why not?), does it retain biological activity or is it fragmented? These PK/ PD data need to be validated with conventional methods, such as ELISA or 125I and activity assays.

Response: Thanks for the comments. Firstly, the interaction between phthalocyanines (including NPTAT) and proteins (including BSA) were thoroughly studied since 1990s, and strong affinity between them had been demonstrated extensively. Many researchers have exploited proteins as phthalocyanines carriers for PDT drugs (such as Dennis K.P. Ng, et al. Journal of Inorganic Biochemistry, 2006, 100, 946–951. K. Lang, et al. Coordination Chemistry Reviews, 2004, 248, 321–350. Xiaolei Zhou, et al. Acta Biomaterialia, 2015, 23, 116–126). On the other hand, researchers had found that covalent bind might induce deactivation of the protein drugs due to the change of their spatial structure or blockage of their active site (Berg JM, et al. Biochemistry. 5th edition. New York: W H Freeman; 2002. Section 8.5, Enzymes Can Be Inhibited by Specific Molecules.).

The *in vitro* release rates of BSA and NPTAT were quantified and added in the revised manuscript as Table 1. We also added the following description in the revised manuscript on line 201, page 7: “The release rates of the BSA and NPTAT were shown in Table 1. The release rate of BSA was ~ 500 times higher than that of NPTAT, which is exactly equal to the mass ratio of BSA and NPTAT in the micro-carriers (500:1), indicating NPTAT can be exploited as excellent tracer for BSA.”

We also have measured the release profile (Supplementary Fig. 14) and quantified the *in vitro* release rates (Supplementary Table 3) of PEP in the revised manuscript.

Furthermore, we have exploited the ¹²⁵I labelling method to investigate the PK of released BSA in the revised manuscript. We found that part of released BSA can be absorbed by the GI tract into the blood stream, their concentration reached to nearly 1.7% ID/g at 6 h after oral administration, while the rest of the absorbed BSA accumulated in internal organs such as liver and kidney (~2% ID/g for the liver and ~3% ID/g for the kidney). After 72 h of circulation, ~ 25% ID of the BSA were cleaned through the urine. Before measuring radioactivity, trichloroacetic acid precipitation method were used to remove free ¹²⁵I ions and fragmented BSA to make sure all radioactive signals were originated from intact BSA. As the proteins used in this work do not have pharmacological activity (BSA is a model protein while PEP is an adjuvant), the PD data cannot be quantitatively analyzed.

In order to make this point clearer for readers, we have added the Fig. 5b-d and following contents in the revised manuscript to show the pharmacokinetics of ¹²⁵I-BSA loaded micro-carriers on line 235, page 8:

“Besides the biodistribution of the micro-carrier, the pharmacokinetics of the released BSA was also investigated by using the radioisotopic tracing method. As a proof of concept, ¹²⁵I-BSA loaded micro-carriers were fabricated and orally administrated to mice. Radioisotopic tracing results from plasma, organs (liver, kidney, stomach and intestine) and excretion (including urine and excrement) showed that part of released BSA can be absorbed by the GI tract into the blood stream,

reaching to nearly 1.7% ID/g at 6 h after oral administration, while the rest of absorbed BSA accumulated in internal organs such as liver and kidney (~ 2% ID/g for the liver and ~ 3% ID/g for the kidney). After 72 h of circulation, ~ 25% ID of the BSA were cleared through the urine (Fig. 5b-d)”

We also have added the following contents in the revised supplementary information on line 11, page 27:

“In vivo pharmacokinetics investigation of BSA by using radioisotopic tracing method.

1000 nm micro-carriers were used in the following experiment.

Labeling and Purification of ^{125}I -BSA

BSA was radiolabeled with ^{125}I -Na using iodogen method. Labeled BSA was separated using Sephadex G-25 (0.4 × 8 cm) column and drenched with 0.01 M phosphate-buffered saline (pH 7.4). The elution fractions were collected and free ^{125}I in the fraction was removed using ultrafiltration tubes with a molecular weight cutoff of 1 kDa. The purity of ^{125}I -BSA was assessed using HPLC method and SDS-PAGE. The radioactivity was measured using γ -counter. The resulting radiochemical purity and activity concentration of the resulting ^{125}I -BSA was measured to be 98% and 1.54 $\mu\text{Ci } \mu\text{g}^{-1}$.

Fabrication of ^{125}I -BSA loaded micro-carriers

0.7 mg ^{125}I -BSA was mixed with 4.3 mg un-labeled BSA in 2 ml MES buffer (0.1 M, pH 5), followed by 15 min stirring at room temperature. 12 mg $\text{SiO}_2\text{-Nd@SiO}_2\text{@mSiO}_2\text{-NH}_2$ was soaked into this solution, followed by stirring at room temperature for 1 h. The as-prepared ^{125}I -BSA-NPTAT loaded $\text{SiO}_2\text{-Nd@SiO}_2\text{@mSiO}_2\text{-NH}_2$ was collected by centrifugation (3000 rpm) and redispersed in 2 ml MES buffer. Then, 3 ml of above SSPI solution (4 mg ml^{-1} in 0.1 M MES buffer) was added quickly, and the obtained turbid solution was further stirred for 1 hours at room temperature. The resulting micro-carriers were finally centrifuged at 3000 rpm, washed several times with MES buffer to remove unreacted SSPI. By measuring the radiation dose of the supernatants, loading efficiency was calculated to be 18.2%.

In vivo pharmacokinetics investigation

Kunming mice (5-6 weeks, female) were fasted for 12 h before experiment. Then the animals were randomized into four groups. Mice in all groups received a single dosage of ^{125}I -BSA loaded micro-carriers (50 mg kg^{-1}) via oral gavaging. The animals were sacrificed at 2, 6, 12 or 72 h after gavaging. Blood, Excretion (including urine and feces) and tissues (including stomach, intestine, liver and kidney) were collected and weighed. Before measuring radioactivity, trichloroacetic acid precipitation method were used to remove free ^{125}I ions and fragmented BSA.”

It is noted that our NIR bioimaging technique (ACIE method) is different from the radioisotopic tracing method. ACIE method can give the information of the release amount of BSA from micro-carriers in real time, but it cannot show the distribution of the released BSA. The biodistribution of release proteins can be detected by radioisotopic tracing method. In order to further compare the BSA biodistribution results measured by ACIE and radioisotopic tracing method, we have listed the detailed results as following tables. Therefore, as we described in the “Discussion” part of the manuscript on line 357, page 12: “Compared with the traditional radioisotopic method, the ACIE NIR bioimaging technique is non-invasive for *in vivo* detection. But only in situ drug release information can be reflected by the ACIE, other techniques such as radioisotopic method are still dependent to track the trace and activity of the released drugs. To sum

up, we expect ACIE technique would be strongly complementary to the traditional radioisotopic tracing method to realize non-invasive semi-quantitative monitoring of drug release.”

NIR bioimaging method		
Time (h)	BSA remaining in the micro-carriers in the GI tract (% ID)	Cumulative release amount of BSA (% ID)
2	78.4 ± 8.8	9.4 ± 2.4
6	41.8 ± 4.4	38.4 ± 5.0
24	20.7 ± 4.8	45.7 ± 5.3
72	4.1 ± 2.8	48.1 ± 4.9

¹²⁵ I method (%ID)				
Time (h)	GI tract (including stomach and intestine) (%ID)	Plasma (%ID)	Liver (%ID)	Kidney (%ID)
2	87.8 ± 12.1	0.7 ± 0.2	1.2 ± 0.3	0.46 ± 0.16
6	40.8 ± 10.2	0.9 ± 0.2	1.3 ± 0.4	0.40 ± 0.10
24	11.7 ± 3.8	0.05 ± 0.02	0.14 ± 0.08	0.02 ± 0.01
72	0.6 ± 0.2	0.02 ± 0.01	0.04 ± 0.02	0.01 ± 0.00

2. Please clarify the goals of this system: is the intent of this system to deliver drugs to the bloodstream or just to the intestine? What are the potential mechanisms by which the particles could transfer from the intestine into the blood or otherwise access the organs? What are the likely transport rates via these mechanisms?

Response: Thanks for the comments. Our goal is to deliver drugs to the intestine, while minimize the health risks by clearance of micro-carriers through the GI tract. In the revised manuscript, we have added Fig. 5b-d and Table 2 to illustrate the PK of BSA. These PK data showed that the released BSA can be absorbed to the bloodstream and accumulated in the organs like liver and kidney, followed by the clearance from the urine. But for the micro-carriers, our PK data (Fig. 4) demonstrated that little deposition of these particles was found in liver, spleen and kidney in all experimental groups, confirming these particles would not across the mucosa and cause adverse health effects to internal environment.

3. There is very little discussion of the data, short-comings or comparisons with current or competing methods in development. In addition to these items, please discuss possible sources of toxicity of these microcarriers, transfer from the intestine to the blood, effects of microparticle size and considerations for using different proteins, etc.

Response: Thanks for the comments. We have added the following contents in the discussion part of revised manuscript on line 342, page 12:

“Micro-carriers provide many attractive features for oral drug delivery, such as high loading capacity, effective protection and high therapeutic index. Moreover, the properties of microcarriers such as tunable pore size (2.5-12.0 nm, supplementary Fig. 16, 17), aspect ratio (2:1-10:1, supplementary Fig. 18), surface charge (negative or positive) and surface hydrophobicity endow them great potential in the field of drug delivery. Moreover, we found our microcarriers have an extended retention time up to 72 h with little deposition in the inner organs, such as liver, spleen and kidney, confirming these micro-carriers would not cross the mucosa and cause adverse health effects to internal environment. And we also noticed that mice in our experiments did not show any signs of discomfort and their weight gain was normal (average weight increase ~ 200% after three months). Meanwhile, after seven days of daily receiving micro-carriers, histopathological examination showed that the morphology of intestinal tissue and tight junctions were intact without any signs of degeneration (Fig. 5e-f), further indicating little healthy risky was caused by the micro-carriers. The extended retention time in GI tract with little toxic property is supposed to be a solution to “medication non-adherence”, which is a major barrier to effective clinical care. (Traverso, G. & Langer, R. Special delivery for the gut. *Nature* **519**, S19-S19 (2015))

The ACIE technique reported here is a NIR bioimaging technique based on the multi-excitation properties of the NIR-II DCNPs (supplementary Fig. 19). Compared with the traditional radioisotopic method, the ACIE NIR bioimaging technique is non-invasive for *in vivo* detection. But only in situ drug release information can be reflected by the ACIE, other techniques such as radioisotopic method are still dependent to track the trace and activity of the released drugs. To sum up, we expect ACIE technique would be strongly complementary to the traditional radioisotopic tracing method to realize non-invasive semi-quantitative monitoring of drug release. In addition, we believe this ACIE technique not only hold great potential for monitoring the oral drug delivery process, but also will propel the development of in situ NIR sensing of biomarkers such as DNA, redox species or specific enzymes. In principle, ACIE technique will expand the library of NIR sensors, because many commercial visible light reporters can also be used for NIR sensing based on the absorption competition induced emission technique.”

Detailed comments:

4. According to Fig. 4a, at 0.25h, the NIR signal in 1000 nm particles group is higher than the other groups, especially 25nm group. It seems like the original amounts of DCNPs in each groups are different. Thus, the larger the particles, the more the DCNPs will be in the particles. The conclusion that larger particles can stay longer in GI tract than smaller particles could not be made without standardizing NIR signal. It will be better to standardize the NIR signal in each groups with relative DCNPs amount.

Response: Thanks for the useful comment. We have already **standardized** these NIR signals for comparison, as shown in supplementary Fig. 9.

5. In Fig. 5g, it will be better to choose PEP loaded micro-carriers without SSPI covered + probe to be control. This could prove it is because SSPI that retain PEP in micro-carriers until releasing

in intestine. This control could prove the higher FL intensity in PEP loaded micro-carriers+probe group is not because micro-carriers itself.

Response: Thanks for this comment. We have added PEP loaded micro-carriers without SSPI covered + probe as control. As shown in **Supplementary Fig. 15**, this control only behavior 2.5 times increase of the FL intensity, which is much lower than the SSPI covered group (~8 times), confirming the SSPI play an important role in the PEP protection.

In the revised manuscript, we have added the following contents on line 334, page 12: “In comparison, control groups including native PEP and micro-carriers without SSPI coating showed little activity because of the PEP degradation in the GI tract.”

We have added Supplementary Fig. 15 with the following figure caption: “*In vivo* bioimaging of mice after gavaged with PEP-NPTAT loaded micro-carriers without SSPI covered and peptide probe. 405 nm was used as excitation source.”

6. What is the size of the pores in the microparticles that allow it to load protein drug? Can these sizes be tunes or will pore size limit the choice of protein drug to load? What are the expected effects of aspect ratio, surface charge and surface hydrophobicity? (for instance, an antibody has a hydrodynamic radius of ~ 10 nm). K_A and k_q and equations (1-4) should be explained more clearly.

Response: Thanks for this comment. Actually, the pore size on the micro-carriers can be easily tuned by changing the reaction condition. In the revised manuscript, we have added the results of micro-carriers with different pore sizes (2.5, 7.7, 8.9 and 12.0 nm, **Supplementary Fig. 16, 17**) and aspect ratios (2:1, 6:1 and 10:1, **Supplementary Fig. 18**). The surface charge can be tuned by post-grafting some amino-modified silica source, such as APTES used in our work (**Supplementary Fig. 1**), while the changing of surface hydrophobicity can be realized by post-grafting some organic silica source (such as 1,2-bis(triethoxysilyl)-ethane (BTEE)) according to previous reports (Xiaomin Li, et al. J. Am. Chem. Soc. 2014, 136, 15086–15092. Xiaomin Li, et al. J. Am. Chem. Soc. 2015, 137, 5903–5906. Yong Wei, et al. Sci. Rep. 2016, 6, 20769.). Obviously, the mesoporous micro-carrier is a good candidate for different kinds of drugs, such as antibody and hydrophobic drugs (even the molecular size is more than 10 nm), etc.

K_A is the binding constant, which reflect the degree of interactions between the NPTAT and BSA, while k_q is the bimolecular quenching constant between them. In order to make the explanation for K_A , k_q and eq (3-5) (equal to eq (2-4) in the original version) clearer for readers, we have added the following contents in the revised manuscript on line 171, page 6:

“Interaction between phthalocyanines and proteins have been widely investigated using FL spectroscopy method. By evaluating the FL quenching effect of proteins after adding a certain amount of NPTAT, the binding constant (K_A) and bimolecular quenching constant (k_q) between them can be quantified by the Stern-Volmer equation (see methods section, Eq. (3-5)). K_A reflect the degree of interactions between the NPTAT and proteins, while k_q can be used to identify the quenching process (static or dynamic quenching).”

And we also added the following contents to explain eq (1) in the revised manuscript on line 254, page 9:

“The intensity of NIR-II signals excited by 808 nm laser were related to the quantity of the micro-carriers in real time while those excited by 730 nm laser were related to the drug release properties. Therefore, a semi-quantitative analysis is conducted to demonstrate the release percentage ($Q_{(\tau)}$) of micro-carriers in real time:

$$Q_{(\tau)} = \frac{F_{730,(\tau)} - F_{730,0}}{F_{730,unloaded} - F_{730,0}} \times 100\% = \frac{F_{730,(\tau)} - F_{730,0}}{\alpha F_{808,(\tau)} - F_{730,0}} \times 100\% \quad (1)$$

Where $F_{730,0}$ is the initial intensity of NIR-II signals under 730 nm excitation for the drug loaded micro-carriers just after gavaging, and $F_{730,(\tau)}$ is used to represent the intensity of NIR-II signals for the drug loaded micro-carriers under 730 nm excitation at a specific time point (τ). $F_{730,unloaded}$ is used to represent the intensity of NIR-II signals for the unloaded micro-carriers under 730 nm excitation. Since the intensity of NIR-II signals under 730 nm excitation will be quenched gradually with the increase of the drug loading amount, $F_{730,unloaded}$ can be used to reflect the total amount of loaded BSA. However, it's difficult to obtain the $F_{730,unloaded}$ in real time because of the quenching effect of the loaded drug, the $F_{730,unloaded}$ can be related to $F_{808,(\tau)}$ directly. Because the NIR-II signals from DCNPs under 730 nm and 808 nm excitation will maintain a fixed ratio when the power density is below 4.55 W cm^{-2} (Supplementary Fig. 10, 11 and Supplementary Table 2). Therefore, $F_{730,unloaded}$ can be expressed as $\alpha F_{808,(\tau)}$, where coefficient α is the constant to represent the $F_{730,unloaded}/F_{808,(\tau)}$, which is measured to be 0.62. Furthermore, since the micro-carrier will be cleaned out with time, $F_{808,(\tau)}$ will change gradually during the drug delivery process. So $F_{808,(\tau)}$ can be used to reflect the amount of unloaded micro-carriers in real time. Taking drug release amount calculation at 12 h after gavaging as an example, the intensity of NIR-II signals under 730 nm ($F_{730,(12 \text{ h})}$) or 808 nm ($F_{808,(12 \text{ h})}$) excitation were calculated to be 23.5 and 75.2, respectively (Fig. 6a). And the $F_{730,0}$ was measured to be 3.0, so the release percentage $Q_{(12 \text{ h})}$ can be estimated to be 47%. As shown in Fig. 6c, the calculated Q was only 10% at the first 2 h, then it showed a burst enhancement in the following 12 h and reached to $\sim 62\%$ at 72 h, suggesting the efficient release behavior of the micro-carriers *in vivo*.”

We have added the following figures in the revised manuscript:

“Supplementary Fig. 16. TEM images of micro-carriers with pore size of 2.5 nm (a), 7.7 nm (b), 8.9 nm (c) and 12.0 nm (d). The pore size can be easily tuned by changing the reaction condition, such as changing the oil used in the synthesis to octadecene (a), decahydronaphthalene (b) and cyclohexane (c). Micro-carriers with pore size of 12 nm (d) were synthesized by reducing the TEOS amount by 25% compared with (c).”

“Supplementary Fig. 17. Nitrogen adsorption-desorption isotherms (a) and pore size distribution (b) of different micro-carriers, respectively. The corresponding BET surface area was measured to be 586.4, 467.3, 366.8 and 330.1 $\text{m}^2 \text{g}^{-1}$, respectively.”

“Supplementary Fig. 18. TEM images of micro-carriers with aspect ratio of 2:1 (a), 6:1 (b) and 10:1 (c).”

7. The use of mice versus rats should be clarified in the results (lines 207 and 228, 358)

Response: We appreciate this advice. We have changed rats to mice in the revised manuscript.

Point-by-Point Response to Referees

Reviewers' comments:

Reviewer #1 (Remarks to the Author):

The manuscript titled 'In-vivo gastrointestinal drug release monitoring through second near-infrared window fluorescent bioimaging with orally delivered micro-carrier' reports a novel microcarrier system of lanthanide-based downconversion nanoparticles and silica particles for bioimaging of gastrointestinal drug release using second near infrared window fluorescent signal. Wang et al., demonstrate that these microcarrier system show a 72 h residence time in the gastrointestinal tract and can be used for bioimaging by exploiting the NIR-II signals from two different excitation sources.

Overall this is an interesting study that evaluates image-guided oral delivery using microparticle systems. Several major issues need to be addressed prior to publication.

Comments:

1. The authors indicate that the microcarrier system has no toxic effects based on the normal weight gain and no signs of discomfort. However, further studies such as the effect on intestinal tissue morphology and tight junctions would be required as a proof of concept to indicating no toxic effects of this microcarrier system on the gastrointestinal tract.

Response: Thanks so much for the useful comments and suggestions. In the revised manuscript, histopathological examination of the intestine was carried out to check the tissue morphology and intestinal tight junctions after oral administration of micro-carriers. All of them were found to be intact without any signs of degeneration, indicating that micro-carriers did not induce damage in these tissues.

We have added the following contents in the revised manuscript on line 366, page 13:

“Meanwhile, after seven days of daily receiving micro-carriers, histopathological examination showed that the morphology of intestinal tissue and tight junctions were intact without any signs of degeneration (Fig. 5e-f), further indicating little healthy risky was caused by the micro-carriers.”

And we have added **Fig. 5e-f** in the revised manuscript with the following figure caption “H&E-stained intestine sections of mice after seven days of daily receiving (e) micro-carriers (50 mg kg⁻¹) and (f) saline water. All treatments were given via a daily oral gavage. Mean ± s.d. for n = 3.”

2. Quantitative biodistribution analysis should be carried out to correlate with qualitative imaging studies.

Response: Thanks for the comments. We have added Fig. 5a to illustrate the correlation of the biodistribution analysis between inductively coupled plasma atomic emission spectrometry (ICP) and NIR imaging measurement. By comparing the percentage of dose results measured by ICP and

NIR bioimaging techniques, we found that the differences between them were not significant ($p > 0.05$) (Fig. 5a), demonstrating the NIR-II bioimaging approach is rather convincing for *in vivo* monitoring the biodistribution of the micro-carriers.

We have added the following contents in the revised manuscript on line 227, page 8:

“By comparing the results measured by ICP and NIR bioimaging techniques, we found that the differences between them were not significant ($p > 0.05$) (Fig. 5a), demonstrating the NIR-II bioimaging approach is rather convincing for *in vivo* monitoring the biodistribution of the micro-carriers.”.

3. Authors need to mention the sample size (n) number in the experimental methodology section wherever applicable.

Response: We appreciate the reviewer for the useful suggestion. In the revised manuscript and supplementary information, we have added the detailed sample size (n) information in the methodology section as following:

We have added “n = 3” in the revised manuscript on line 417, page 14.

We have added “n = 3 per group” in the revised manuscript on line 435, 440, page 15 and line 449, 459, page 16.

We have added “mean \pm s.d. for n = 3” in the figure caption of Fig. 3d, Fig. 4, Fig. 5 and Fig. 6.

We have added “Representative images for n = 3 per group” in the figure caption of Fig. 3, Fig. 4a and Fig. 6a,d,g.

We have added “with n = 3 per group” in the revised supplementary information on line 542, page 30.

4. With respect to characterization of this microcarrier system, it is essential to determine and discuss the maximum loading ability of this microcarrier system with respect to a model proteins used in the study.

Response: Thanks so much for the comment. We have added the maximum loading capacity in the revised manuscript as following: “maximum loading capacity for BSA is 47.4 wt. %” (on line 182, page 6) and “maximum loading capacity for PEP is 40.4 wt. %” (on line 304, page 11).

In a previous work reported by us (Shen, D., *et al.*, *Nano Lett.* **14**, 923-932 (2014).), we have discussed whether the different pore sizes can affect the loading and releasing amounts of protein drugs in detail. With bovine β -lactoglobulin (~ 5 nm) as a model protein, it was found that the micro-carriers with 5.5 nm pore size have the protein loading capacity of ~ 30.5 wt. %, while the micro-carriers with a larger mesopore size of 10 nm show a double loading capacity as 62.1 wt. %. Therefore, we can conclude that the pore size determines the loading capacity.

The pore size on the micro-carriers can be easily tuned by changing the reaction condition. In the revised manuscript, we have added the results of micro-carriers with different pore sizes (2.5, 7.7, 8.9 and 12.0 nm, **Supplementary Fig. 18, 19**). Therefore, the tunable pore sizes on the micro-carriers will make a lot of protein drugs available to load.

5. The y-axis for Figure 5 b, e and h needs to be well defined.

Response: Thanks for the comments. Fig. 5 has been renamed as Fig. 6 in the revised manuscript. The signal intensities in Figure 6b,e and h have already been normalized. In the revised manuscript, we have renamed y-axis of Figure 6b,e with “NIR signal intensity (a.u.)”, while y-axis of Figure 6h has been renamed with “FL signal intensity (a.u.)”. And we also provided the normalized intensity number in y-axis of these revised figures.

6. The amount of protein/drug delivered at the desired location is an important factor in evaluating the efficacy of the formulation as a carrier system. The authors demonstrate that the microcarrier system show 65% release of its contents in the intestinal sections at pH 8. Furthermore, the authors have confirmed the protein encapsulated is protected from degradation and retains its activity when released from the carrier system at pH 8. It could be ideal to demonstrate that the amount of (model) protein released from the microcarrier system in the intestinal section at pH 8 is suitable to bring about therapeutic effect.

Response: We appreciate the reviewer for the useful suggestion. The protein drugs vary widely in the therapeutic effective dose. As for PEP (a model protein used in the present work), only 1 µg per mouse (125-200 g) would be enough to bring about therapeutic effect for celiac disease according to previous reports (Fuhrmann, G., et al. *Nat. Chem.* **5**, 582-589 (2013). Fuhrmann, G., et al. *Proc. Natl. Acad. Sci.* **108**, 9032–9037 (2011).). Since we have demonstrated that the cumulative PEP release amount can reach to 54.8 ± 10.2 µg per mouse (20-30 g) within 6 h, followed by a sustained increase to 125.7 ± 11.8 µg per mouse (20-30 g) at 24 h (Supplementary Table 4), the released amount of PEP from micro-carriers were considerably suitable to bring about effective treatment. In order to make this point clearer for readers, we added the following description in the revised manuscript on line 324, page 11. We also added Supplementary Table 5 in the revised supplementary information to show the cumulative PEP release amount at different time point.

“Therefore, the PEP release amount can be predicted by using the same equations (Eq. (1) and Eq. (2)) for BSA-NPTAT system. As shown in Fig. 6f and supplementary Table 5, the calculated Q and M_t were 25% and 54.8 ± 10.2 µg respectively at 6 h, then a burst release was observed in the following 18 h and reached to ~ 60% and 125.7 ± 11.8 µg at 24 h, respectively. According to previous reports, only 1 µg of PEP in the GI tract would be enough to bring about therapeutic effect for a diseased mouse. Therefore, the drug delivery capacity for PEP with micro-carriers was high enough for the effective treatment.”

To further demonstrate the exact amount of BSA released from the micro-carriers, we have added the following contents in the revised manuscript on line 279, page 10:

“Based on the obtained release percentage $Q_{(\tau)}$, cumulative release amount of BSA in real-time can be derived by the following equation:

$$M_{(\tau)} = M_{(\tau-1)} + \frac{F_{808,(\tau-1)} + F_{808,(\tau)}}{2F_{808,0}} m\theta (Q_{(\tau)} - Q_{(\tau-1)}) \quad (2).$$

Where $M_{(\tau)}$ and $M_{(\tau-1)}$ are the cumulative release amounts of BSA at two closed time point (τ and $\tau-1$). $Q_{(\tau)} - Q_{(\tau-1)}$ reflects the release percentage during this time period (from time ($\tau-1$) to time (τ)). $F_{808,(\tau-1)}$, $F_{808,(\tau)}$ and $F_{808,0}$ are used to represent the intensity of NIR-II signals under 808 nm excitation for drug loaded micro-carriers at time ($\tau-1$), time (τ) and initial time point just after gavaging. Since the *in vivo* delivery of micro-carriers is a dynamic process, the micro-carriers will be cleaned out gradually, the real-time micro-carrier percentage in the body can be calculated from

$\frac{F_{808,(\tau-1)}+F_{808,(\tau)}}{2F_{808,0}}$. Coefficient m and θ is the initial micro-carriers mass and drug loading percentage, respectively. Therefore, the release amount of BSA from micro-carriers during a specific period (from time $(\tau-1)$ to time (τ)) can be calculated from $\frac{F_{808,(\tau-1)}+F_{808,(\tau)}}{2F_{808,0}}m\theta(Q_{(\tau)} - Q_{(\tau-1)})$. Taking the cumulative drug release calculation from 6 h to 12 h as an example, $Q_{(12\text{ h})}$ and $Q_{(6\text{ h})}$ value were calculated to be 48% and 42% according to Eq. (1), respectively (Fig. 6c). Meanwhile, $F_{808,(6\text{ h})}$, $F_{808,(12\text{ h})}$ and $F_{808,0}$ value were measured to be 83.5, 75.2 and 93.3 (derived from Fig. 6b), so the $M_{(12\text{ h})}$ was estimated to be $103.3 \pm 5.7 \mu\text{g}$. The calculated cumulative drug release amounts were showed in Table 2. It was shown that the cumulative release amount of BSA reached to $108.1 \pm 12.3 \mu\text{g}$ at 24 h, while it merely increased to $117.3 \pm 11.3 \mu\text{g}$ at 72 h, indicating the burst release at first 24 h, followed by a sustained release behavior. These findings match well with the release profile of *in vitro* experiment, and also prove the feasibility of our ACIE method for the *in vivo* drug release measuring in real-time.”

7. Language errors need to be corrected.

Response: Thanks so much for the reviewer’s comments. We have polished our manuscript carefully and corrected the grammatical, styling, and mistyped errors found in our manuscript. Detailed modifications are also provided as following.

We corrected the following mistake on page 3, line 75: “drug release content” was changed to “drug release behavior”.

We corrected the following mistake on page 3, line 80: “a key constraint” was changed to “key constraints”.

We corrected the following mistake on page 4, line 109: “can be” was changed to “were”.

We corrected the following mistake on page 4, line 118: “It was estimated ~ 200 DCNPs were attached on each SiO₂ particle according to the inductively coupled plasma atomic emission spectrometry (ICP) measurement.” was changed to “According to the inductively coupled plasma atomic emission spectrometry (ICP) measurement, ~ 200 DCNPs were attached on each SiO₂ particle.”

We corrected the following mistake on page 5, line 143: “mSiO₂ shells” was changed to “mSiO₂ shell”.

We corrected the following mistake on page 5, line 148: “acceptor” was changed to “acceptors”.

We corrected the following mistake on page 5, line 149: “overlapping absorption spectra with” was changed to “absorption overlapping with”.

We corrected the following mistake on page 5, line 152: “NPTAT organic dyes with a maximum absorption wavelength of 625 nm” was changed to “NPTAT organic dye with a maximum absorption at 625 nm”

We corrected the following mistake on page 6, line 156: “not any” was changed to “none”.

We corrected the following mistake on page 6, line 158: “could be” was changed to “was”

We corrected the following mistake on page 6, line 166: “unconspicuous” was changed to “inconspicuous”.

We corrected the following mistake on page 7, line 202: “extending” was changed to “extended”.

We corrected the following mistake on page 7, line 203: “is also correlated well the spectroscopy

results” was changed to “were also correlated well with the spectroscopy results”.

We corrected the following mistake on page 8, line 225: “microcarrier” was changed to “micro-carrier”.

We corrected the following mistake on page 11, line 306: “are” was changed to “were”.

We corrected the following mistake on page 11, line 303 and page 11, line 305: “is” was changed to “was”.

Reviewer #2 (Remarks to the Author):

In this paper, Wang and co-workers fabricated a novel kind of DCNPs loaded mesoporous microparticles as an efficient drug delivery carrier for NIR-II monitored oral drug delivery. Taking advantage of the multi-absorption property of lanthanide ions, a novel absorption competition system has been utilized to predict the delivered content in real time. To my point of view, this system can be seen as a big breakthrough in this optical biosensing research area. It provides a feasible way for monitoring or detecting biomolecule in NIR-II region with deeper penetration depth and higher resolution, compared with those fluorescent probes working at traditional visible or NIR-I region. I therefore strongly recommend its publication in Nature communications. Some minor revisions that could further improve the technical content of this work are outlined below.

Comments:

1. *The release content monitoring technique used in this work actually is semi-quantitative, because the released content and the change of NIR-II signals are linked by the amount of NPTAT, rather than directly feedback. Therefore, I suggest that it will be proper to replace the phrase “quantitative” with “semi-quantitative” in the whole manuscript.*

Response: We accept this comment. We have replaced the phrase “quantitative” with “semi-quantitative” in the revised manuscript on line 252, page 9 and line 376, page 13. And we also have replaced the phrase “quantitatively” with “semi-quantitatively” on line 39, page 2 and line 99, page 4.

2. *The absorption spectrum of DCNPs in Figure 3a is unclear, please provide a clearer image to displace it.*

Response: Thanks for the comment. In order to make the absorption spectrum of DCNPs in Figure 3a more identifiable, we have added the Supplementary Fig. 5 in the revised supplementary information to show the absorption spectrum of DCNPs from 725 nm to 825 nm.

We have added **Supplementary Fig. 5** with the following figure caption:
“Supplementary Fig. 5. Absorption spectrum of DCNPs from 725 nm to 825 nm.”

3. *In Figure 4c, there are some mistakes displayed in the x-axis. “Excrement” is partially shadowed.*

Response: We appreciate this suggestion. We have modified the x-axis of Figure 4c to show the “Excrement” clearly in the revised manuscript.

Technical comments:

1. The emission of Nd doped DCNPs consists of several bands. Please provide the proportion of 1060 nm emission intensity among these bands at different power density.

Response: Thanks for the useful suggestion. In the revised manuscript, we have added **Supplementary Fig. 12** to show three primary emission bands (860, 1060 and 1340 nm) of Nd doped DCNPs under 730 nm or 808 nm excitation. Furthermore, we summarized the proportion of three primary emission bands (860, 1060 and 1340 nm) of Nd doped DCNPs under 730 nm or 808 nm excitation with different power density in the Supplementary Table 3.

Supplementary Fig. 12 and **Supplementary Table 3** have been added in the revised supplementary information with the following figure caption:

“Supplementary Fig. 12. A typical downconversion emission spectrum of NaGdF₄:5%Nd@NaGdF₄ DCNPs under 730-nm or 808-nm excitation.”

“Supplementary Table 3. The proportion of primary emission bands (860, 1060 and 1340 nm) of NaGdF₄:5%Nd@NaGdF₄ DCNPs excited by 730 or 808 nm at different power density”

2. Neodymium ion possesses multi-absorption property. Is there also multi-excitation property?

Response: Thanks for the useful comment. Actually, we found that there are four primary excitation bands (730, 800, 862 and 894 nm) of Neodymium (Nd) ions doped DCNPs in NIR region. All of the excitation focused on these bands can lead to 1060 nm emission. In order to make this point clearer for readers, we have added **Supplementary Fig. 21** in the revised manuscript with the following figure caption:

“Supplementary Fig. 21. Multi-excitation property of the NaGdF₄:5%Nd@NaGdF₄ DCNPs. The NIR emission spectrum of NaGdF₄:5%Nd@NaGdF₄ DCNPs (20 mg ml⁻¹, dispersed in cyclohexane) excited by Xenon lamp at different wavelength.”

3. Whether the NIR cyanine dyes can be used in absorption competition system or not? They have wide varieties and are commercial available with controlled absorption wavelength.

Response: Thanks so much for the comment. As discussed in the manuscript on line 149, page 5, there are some requirements for the absorbing competition acceptor, such as absorption overlapping with DCNPs, little fluorescence and strong affinity with drugs. Obviously, the biggest challenge for commercial NIR cyanine dyes to be used as absorbing competition acceptor (like IR-780 or IR-820) is their strong NIR emission, which would significantly decrease the signal/noise ratio of the DCNPs, as they always share the similar excitation bands with lanthanide doped DCNPs. Therefore, we don't suggest to use NIR cyanine dyes as absorbing competition acceptor. The related explain has been shown in the manuscript on line 155, page 6 as following: “Moreover, there is none fluorescence (FL) from NPTAT to interfere with NIR-II signals of DCNPs under 730 or 808 nm excitation.”

4. Why is the long retention time in GI tract helpful in oral drug delivery? Please explain it.

Response: We appreciate this suggestion. Generally, long retention behavior of the drug carrier can be exploited as prolonged drug delivery system, which has more flexibility in dosage design than the conventional drug delivery system. For example, in 2015, Robert Langer, et al. reported an elastic, pH-responsive supramolecular gel as gastric-resident devices, which were considered as good oral drug delivery carrier due to their prolonged gastric retention time (two to seven days) (Robert Langer, et al. *Nature Materials* **14**, 1065–1071 (2015)).

Our pharmacokinetics results (ICP results, Fig. 4b-g) showed that large particles such as 1000 nm micro-carriers have longer retention time (72 h) in the GI tract than the smaller ones like 25, 80 and 300 nm micro-carriers. While less than 0.1% of the 1000 nm micro-carriers were absorbed into the viscera organs (such as liver, spleen and kidney) during the whole process, further demonstrating the 1000 nm micro-carrier can serve as good prolonged drug delivery carrier with little toxicity.

In order to make this point clearer for readers, we have added the following contents in the revised manuscript on line 362, page 13:

“Long retention behavior of the drug carrier can be exploited as prolonged drug delivery system, which has more flexibility in dosage design than the conventional drug delivery system (Robert Langer, et al. *Nature Materials* **14**, 1065–1071 (2015)).”

5. Why these particles cannot be absorbed by the body? Besides the particle size, I think they might be affected by the surface properties.

Response: Thanks so much for the comment. Actually, it can be observed that the extent of visceral organ uptake for the micro-carriers through the GI tract is size dependent according to the visceral organ (liver, spleen and kidney) distribution results shown in Fig. 4b-g. The highest uptake amount (including liver, spleen and kidney) was reached at 12 h after gavaging (1.15, 0.58 and 0.41% for 25, 80 and 300 nm micro-carriers). In contrast, the uptake amount of 1000 nm micro-carriers was below 0.1% throughout the oral drug delivery process, indicating minor healthy risk would be caused by the 1000 nm micro-carriers.

Previous reports had demonstrated that absorption of particles predominantly took place at the intestinal lymphatic tissues (i.e., Peyer’s patches) (A. Hillery, P. Jani, A. Florence, *J. Drug Targeting* **2**, 151–156 (1994). M.E. LeFevre, A.M. Boccio, D.D. Joel, *Pro. Soc. Exp. Biol. Med.* **190**, 23–27 (1989)). The epithelial cell layer overlying the Peyer’s patches contains specialized M cells which are believed to be transcytotic (J. Pappo, T.H. Ermak, *Clin. Exp. Immunol.*, **76**, 144–148 (1989)). Aside from particle size, previous reports also demonstrated that the particle absorption rate as well as absorption efficiency by Peyer’s patches was affected by the surface properties of the particles, such as surface charge and hydrophobicity.

Generally, hydrophobic particles were absorbed more readily than hydrophilic ones (J.H. Eldridge, C.J. Hammond, J.A. Meulbroek, J.K. Staas, R.M. Gilley, T.R. Tice, *J. Controlled Release*, **11**, 205–214 (1990)). For example, S.S. Davis and L. Illum have shown that the uptake of polystyrene microparticles was decreased by weakening the surface hydrophobicity of particles through the adsorption of poloxamers or poloxamines (S.M. Moghimi, C.J. Porter, I.S. Muir, L. Illum, S.S. Davis, *Biochem. Biophys. Res. Comm.*, **177**, 861–866 (1991) M.E. Norman, P. Williams, L. Illum, *Biomaterials*, **14**, 193–202 (1993)).

According to previous reports, the surface charge of particles also can affect their absorption

rate and efficiency through changing their mucoadhesive properties. For example, T. Florence et al. demonstrated that carboxylated poly(styrene) particle (negative surface charge) show significantly decreased affinity to intestinal epithelia with negative surface charge, especially to M cells, compared to poly(styrene) particle with positive surface charge (P. Jani, G.W. Halbert, J. Langridge, T. Florence, *J. Pharm. Pharmacol.*, **41**, 809-812 (1989)).

In summary, the surface properties of our micro-carrier in the intestine is hydrophilic and negatively charged (- 15.4 mV) due to the deprotonation of abundant carboxyl group on SSPI. The hydrophilic surface will hinder the absorption of micro-carriers through Peyer's patches, while the negatively charged surface may decrease their affinity with intestinal epithelia, especially to M cells in the Peyer's patches. Therefore, the low absorption property of the 1000 nm micro-carriers should be due to their surface properties with a combination of negatively charged hydrophilic surface and large size.

The uptake transport rate of our micro-carriers is very slow due to their negatively charged hydrophilic surface. In order to make it more clearly, we have summarized the percentage of dose (% D) of different sized micro-carriers in visceral organs (including liver, spleen and kidney) as a function of time:

	25 nm (% D)	80 nm (% D)	300 nm (% D)	1000 nm (% D)
2 h	0.20 ± 0.12	0.12 ± 0.08	0.06 ± 0.03	0.00 ± 0.00
6 h	0.38 ± 0.11	0.23 ± 0.17	0.16 ± 0.05	0.08 ± 0.06
12 h	1.15 ± 0.41	0.58 ± 0.15	0.41 ± 0.18	0.03 ± 0.01
24 h	0.27 ± 0.07	0.20 ± 0.09	0.16 ± 0.07	0.07 ± 0.02
48 h	0.12 ± 0.04	0.07 ± 0.02	0.06 ± 0.01	0.04 ± 0.01
72 h	0.05 ± 0.02	0.05 ± 0.02	0.04 ± 0.01	0.05 ± 0.01

Even for the 25 nm particle, which have the fastest uptake transport rate among different sized micro-carriers, it took 12 h to reach the maximum accumulative amount (1.15 ± 0.41% D) in visceral organs. Therefore, the estimated absorption transport rate of our micro-carrier with different size is below 0.1% D h⁻¹ ($\frac{1.15 \pm 0.41 \% D}{12 h}$).

We have added the following contents in the revised manuscript on line 348, page 12:

“The majority of the available evidence in the literature suggests that absorption of particles predominantly takes place at the intestinal lymphatic tissues (*i.e.*, Peyer’s patches). The epithelial cell layer overlying the Peyer’s patches contains specialized M cells which are believed to be transcytotic. Aside from particle size, the particle absorption rate as well as absorption efficiency by Peyer’s patches has also been shown to be affected by the surface properties of the particles, such as surface charge and hydrophobicity. Generally, hydrophobic particles were absorbed more readily than hydrophilic ones, while positively charged particles will increase the mucoadhesive properties of particles. In this work, we found the retention time and deposition amount in the visceral organs is size dependent. In this work, we found the retention time and deposition amount in the visceral organs is size dependent. It is noted that the smaller micro-carriers, such as 25 nm micro-carriers, showed a relative short retention time (less than 24 h) and high uptake amount (~ 1.15% at 12 h, Fig. 4b-g). Therefore, further toxicity investigation for the small micro-carriers is required before using, because these micro-carriers in the bloodstream may induce hemolysis, allergic reaction and

even some severe diseases. In contrast, 1000 nm micro-carriers used in our protein delivery experiment exhibit many merits as a good oral drug carrier, such as extended retention time up to 72 h and little deposition (< 0.1%) in the visceral organs. Long retention behavior of the drug carrier can be exploited as prolonged drug delivery system, which has more flexibility in dosage design than the conventional drug delivery system. And we also noticed that mice in our experiments did not show any signs of discomfort and their weight gain was normal (average weight increase ~ 200% after three months). Meanwhile, after seven days of daily receiving micro-carriers, histopathological examination showed that the morphology of intestinal tissue and tight junctions were intact without any signs of degeneration (Fig. 5e-f), further indicating little healthy risky was caused by the micro-carriers.”

Reviewer #3 (Remarks to the Author):

This study reports a new micro-carrier to facilitate oral protein drug delivery. This micro-carrier protects drugs during passage through the stomach and allows for tracking of micro-carrier anatomical location via near-IR fluorescence. The particles are made and thoroughly characterized using standard techniques. An NPTAT dye was co-loaded with BSA protein and used as a proxy to monitor protein PK and in vitro release profiles monitored at several pH. Particles of varying size were gavaged into mice and rats and found to clear >80% particles in 24 hours. Finally, a model enzyme PEP was administered with NPAT and a peptide substrate appending with a pair of FRET probes to discriminate between cleaved and uncleaved peptide.

This is an important area of research and a lot of work was performed in this study. The authors present an interesting approach for both micro-carrier design and PK monitoring of the micro-carriers, using lanthanide-based down-conversion nanoparticles with near-IR fluorescence and a mesoporous structure. Non-invasive monitoring via fluorescence is very important and the key advance here is the enhanced bio-imaging, including 1-2 cm tissue penetration and sub-10 um resolution combined with protein drug delivery. However several key questions related to feasibility of their approach for drug delivery were not addressed. In particular, their protein drug PK data is not clear (where did the protein go, how fast and while retaining how much activity) and importantly, was not validated using standard techniques.

Major comments:

1. While the PK of the microcarriers is an important questions, the PK/ PD of the protein drug is at least, if not more important. Despite the title, the protein PK has not really been monitored or reported. Why was the reporting fluorophore not covalently coupled to BSA to be a more robust report of protein location? The in vitro BSA and NPTAT release rates should be quantified and compared in the text as part of the validation (data in Figure 3d; lines 190-200). Since the PEP protein is a different protein with different surface hydrophobicity/ charge characteristics, the in vitro loading and release profiles also need to be measured and reported for this protein. Figure 5, which ostensibly shows the in vivo protein PK data are unclear and unconvincing. How much protein is released, where does it go (is it also restricted to the intestinal lumen? If not, why not?), does it retain biological activity or is it fragmented? These PK/ PD data need to be validated with conventional methods, such as ELISA or 125I and activity assays.

Response: Thanks for the comments. Since there are a lot of questions in the comment 1, we will response them one by one.

Question (1): *Why was the reporting fluorophore not covalently coupled to BSA to be a more robust report of protein location?*

Response (1): Firstly, the non-covalent interaction between phthalocyanines (including NPTAT) and proteins (including BSA) were thoroughly studied since 1990s, and strong affinity between them had been demonstrated extensively. Many researchers have exploited proteins as phthalocyanines carriers for PDT drugs (such as Dennis K.P. Ng, et al. Journal of Inorganic Biochemistry, 2006, 100, 946–951. K. Lang, et al. Coordination Chemistry Reviews, 2004, 248, 321–350. Xiaolei Zhou, et al. Acta Biomaterialia, 2015, 23, 116–126). On the other hand, previous reports had found that covalent bind might induce deactivation of the protein drugs due to the change of their spatial structure or blockage of their active site (Berg JM, et al. Biochemistry. 5th edition. New York: W H Freeman; 2002. Section 8.5, Enzymes Can Be Inhibited by Specific Molecules.).

Question (2): *The in vitro BSA and NPTAT release rates should be quantified and compared in the text as part of the validation (data in Figure 3d; lines 190-200).*

Response (2): The *in vitro* release rates of BSA and NPTAT were quantified and compared in the **Table 1** of revised manuscript. We also added the following description in the revised manuscript on line 196, page 7: “Moreover, as shown in Table 1, the release rate of BSA was always ~ 500 times higher than that of NPTAT, which was exactly equal to the mass ratio of BSA and NPTAT in the micro-carriers (500:1), indicating NPTAT can be exploited as excellent tracer for BSA.”

Question (3): *Since the PEP protein is a different protein with different surface hydrophobicity/charge characteristics, the in vitro loading and release profiles also need to be measured and reported for this protein.*

Response (3): We have provided the release profile (**Supplementary Fig. 16**) and the *in vitro* release rates (**Supplementary Table 3**) of PEP in the revised supplementary information.

Question (4): *Figure 5, which ostensibly shows the in vivo protein PK data are unclear and unconvincing. How much protein is released, where does it go (is it also restricted to the intestinal lumen? If not, why not?), does it retain biological activity or is it fragmented? These PK/ PD data need to be validated with conventional methods, such as ELISA or 125I and activity assays.*

Response (4): We have exploited the ¹²⁵I labelling method to investigate the PK of released BSA in the revised manuscript. We found that part of released BSA can be absorbed by the GI tract into the blood stream, their concentration reached to nearly 1.7% D g⁻¹ at 6 h after oral administration, while the rest of the absorbed BSA accumulated in internal organs such as liver and kidney (~2% D g⁻¹ for the liver and ~3% D g⁻¹ for the kidney). After 72 h of circulation, ~ 25% of the BSA were cleaned through the urine. Before measuring radioactivity, trichloroacetic acid precipitation method were used to remove free ¹²⁵I ions and fragmented BSA to make sure all radioactive signals were originated from intact BSA. As the proteins used in this work do not have pharmacological activity (BSA is a model protein while PEP is an adjuvant), the PD data cannot be quantitatively analyzed.

In order to make this point clearer for readers, we have added the Fig. 5b-d and following contents in the revised manuscript to show the pharmacokinetics of ¹²⁵I-BSA loaded micro-carriers

on line 231, page 8:

“Besides the biodistribution of the micro-carriers, the pharmacokinetics of the released BSA was also investigated by using the radioisotopic tracing method. As a proof of concept, ^{125}I -BSA loaded 1000 nm micro-carriers were prepared and orally administrated to mice. Radioisotopic tracing results from plasma, organs (liver, kidney, stomach and intestine) and excretion (including urine and excrement) showed that part of released BSA can be absorbed by the GI tract into the blood stream, reaching to nearly 1.7% D g^{-1} (% D g^{-1} is short for percentage of dose per gram) in the plasma at 6 h after oral administration, while the rest of absorbed BSA accumulated in visceral organs such as liver and kidney ($\sim 2\%$ D g^{-1} for the liver and $\sim 3\%$ D g^{-1} for the kidney). After 72 h of circulation, $\sim 25\%$ of the BSA were cleaned out of body through the urine (Fig. 5b-d).”

We also have added the following contents in the revised supplementary information on line 520, page 29:

“In vivo pharmacokinetics investigation of BSA by using radioisotopic tracing method

1000 nm micro-carriers were used in the following experiment.

Labeling and Purification of ^{125}I -BSA

BSA was radiolabeled with ^{125}I -Na using iodogen method. Labeled BSA was separated using Sephadex G-25 (0.4 × 8 cm) column and drenched with 0.01 M phosphate-buffered saline (pH 7.4). The elution fractions were collected and free ^{125}I in the fraction was removed using ultrafiltration tubes with a molecular weight cutoff of 1 kDa. The purity of ^{125}I -BSA was assessed using HPLC method and SDS-PAGE. The radioactivity was measured using γ -counter. The resulting radiochemical purity and activity concentration of the resulting ^{125}I -BSA was measured to be 98% and 1.54 $\mu\text{Ci } \mu\text{g}^{-1}$.

Fabrication of ^{125}I -BSA loaded micro-carriers

0.7 mg ^{125}I -BSA was mixed with 4.3 mg un-labeled BSA in 2 ml MES buffer (0.1 M, pH 5), followed by 15 min stirring at room temperature. 12 mg $\text{SiO}_2\text{-Nd@SiO}_2\text{@mSiO}_2\text{-NH}_2$ was soaked into this solution, followed by stirring at room temperature for 1 h. The as-prepared ^{125}I -BSA-NPTAT loaded $\text{SiO}_2\text{-Nd@SiO}_2\text{@mSiO}_2\text{-NH}_2$ was collected by centrifugation (3000 rpm) and redispersed in 2 ml MES buffer. Then, 3 ml of above SSPI solution (4 mg ml^{-1} in 0.1 M MES buffer) was added quickly, and the obtained turbid solution was further stirred for 1 hours at room temperature. The resulting micro-carriers were finally centrifuged at 3000 rpm and washed several times with MES buffer to remove unreacted SSPI. By measuring the radiation dose of the supernatants, loading efficiency was calculated to be 18.2%.

In vivo pharmacokinetics investigation

Kunming mice (5-6 weeks, female) were fasted for 12 h before experiment. Then the animals were randomized into four groups with $n = 3$ per group. Mice in all groups received a single dosage of ^{125}I -BSA loaded micro-carriers (50 mg kg^{-1}) via oral gavaging. The animals were sacrificed at 2, 6, 12 or 72 h after gavaging. Blood, excretion (including urine and feces) and tissues (including stomach, intestine, liver and kidney) were collected and weighed. Before measuring radioactivity, trichloroacetic acid precipitation method were used to remove free ^{125}I ions and fragmented BSA.”

It is noted that our NIR bioimaging technique (ACIE method) is different from the radioisotopic tracing method. NIR bioimaging can give the information of the release amount of BSA from micro-carriers in real time, but it cannot show the distribution of the released BSA. The biodistribution of released proteins can be detected by radioisotopic tracing method. In order to

further compare the BSA biodistribution results measured by NIR bioimaging and radioisotopic tracing method, we have listed the detailed results in following tables. Therefore, we have added following contents in the “Discussion” part of the manuscript on line 371, page 13: “Compared with the traditional radioisotopic method, the ACIE NIR bioimaging technique is non-invasive for *in vivo* detection. But only in-situ drug release information can be reflected by the ACIE. Other techniques such as radioisotopic method are still needed to track the trace and activity of the released drugs. To sum up, we expect ACIE technique would be strongly complementary to the traditional radioisotopic tracing method to realize non-invasive *in vivo* semi-quantitative monitoring of drug release.”

NIR bioimaging method		
Time (h)	BSA remaining in the micro-carriers in the GI tract (% D)	Cumulative release amount of BSA (% D)
2	78.4 ± 8.8	9.4 ± 2.4
6	41.8 ± 4.4	38.4 ± 5.0

¹²⁵ I method (% D)				
Time (h)	GI tract (% D)	Plasma (% D)	Liver (% D)	Kidney (% D)
2	87.8 ± 12.1	0.7 ± 0.2	1.2 ± 0.3	0.46 ± 0.16
6	40.8 ± 10.2	0.9 ± 0.2	1.3 ± 0.4	0.40 ± 0.10

2. Please clarify the goals of this system: is the intent of this system to deliver drugs to the bloodstream or just to the intestine? What are the potential mechanisms by which the particles could transfer from the intestine into the blood or otherwise access the organs? What are the likely transport rates via these mechanisms?

Response: Thanks for the comments. Since there are a lot of questions in the comment 2, we will response them one by one.

Question (1): Please clarify the goals of this system: is the intent of this system to deliver drugs to the bloodstream or just to the intestine?

Response (1): Our goal is to deliver drugs to the intestine, while minimize the health risks by clearance of micro-carriers through the GI tract. In the revised manuscript, we have added Fig. 5b-d and Table 2 to illustrate the PK of BSA. These PK data showed that the released BSA can be absorbed to the bloodstream and accumulated in the organs like liver and kidney, followed by the clearance from the urine. But we also compared the distribution of different sized micro-carriers in organs like liver, spleen and kidney. It can be observed that the extent of visceral organ uptake for the micro-carriers through the GI tract is size dependent according to the visceral organ (liver, spleen and kidney) distribution results shown in Fig. 4b-g, supplementary Table 2 and supplementary Fig. 10. The highest uptake amount (including liver, spleen and kidney) was reached at 12 h after gavaging (1.15, 0.58 and 0.41% for 25, 80 and 300 nm micro-carriers). In contrast, the uptake amount of 1000 nm micro-carriers was below 0.1% throughout the oral drug delivery process,

indicating minor healthy risky would be caused by the 1000 nm micro-carriers.

Question (2): *What are the potential mechanisms by which the particles could transfer from the intestine into the blood or otherwise access the organs? What are the likely transport rates via these mechanisms?*

Response (2): Previous reports had demonstrated that absorption of particles predominantly took place at the intestinal lymphatic tissues (i.e., Peyer's patches) (A. Hillery, P. Jani, A. Florence, *J. Drug Targeting* **2**, 151–156 (1994). M.E. LeFevre, A.M. Boccio, D.D. Joel, *Pro. Soc. Exp. Biol. Med.* **190**, 23–27 (1989).). The epithelial cell layer overlying the Peyer's patches contains specialized M cells which are believed to be transcytotic (J. Pappo, T.H. Ermak, *Clin. Exp. Immunol.*, **76**, 144–148 (1989).). Aside from particle size, previous reports also demonstrated that the particle absorption rate as well as absorption efficiency by Peyer's patches was affected by the surface properties of the particles, such as surface charge and hydrophobicity.

Generally, hydrophobic particles were absorbed more readily than hydrophilic ones (J.H. Eldridge, C.J. Hammond, J.A. Meulbroek, J.K. Staas, R.M. Gilley, T.R. Tice, *J. Controlled Release*, **11**, 205–214 (1990).). For example, S.S. Davis and L. Illum have shown that the uptake of polystyrene microparticles was decreased by weaken the surface hydrophobicity of particles through the adsorption of poloxamers or poloxamines (S.M. Moghimi, C.J. Porter, I.S. Muir, L. Illum, S.S. Davis, *Biochem. Biophys. Res. Comm.*, **177**, 861–866 (1991) M.E. Norman, P. Williams, L. Illum, *Biomaterials*, **14**, 193–202 (1993).).

According to previous reports, the surface charge of particles also can affect their absorption rate and efficiency through changing their mucoadhesive properties. For example, T. Florence et al. demonstrated that carboxylated poly(styrene) particle (negative surface charge) show significantly decreased affinity to intestinal epithelia with negative surface charge, especially to M cells, compared to poly(styrene) particle with positive surface charge (P. Jani, G.W. Halbert, J. Langridge, T. Florence, *J. Pharm. Pharmacol.*, **41**, 809-812 (1989).).

In summary, the surface properties of our micro-carrier in the intestine is hydrophilic and negatively charged (- 15.4 mV) due to the deprotonation of abundant carboxyl group on SSPI. The hydrophilic surface will hinder the absorption of micro-carriers through Peyer's patches, while the negatively charged surface may decrease their affinity with intestinal epithelia, especially to M cells in the Peyer's patches. Therefore, the low absorption property of the 1000 nm micro-carriers should be attribute to their surface properties with a combination of negatively charged hydrophilic surface and large size.

The uptake transport rate of our micro-carriers by GI tract is very slow due to their negatively charged hydrophilic surface. In order to make it more clearly, we have summarized the uptake percentage of different sized micro-carriers in visceral organs (including liver, spleen and kidney) as a function of time (added as **Supplementary Table 2** in the revised supplementary information).

	25 nm (% D)	80 nm (% D)	300 nm (% D)	1000 nm (% D)
2 h	0.20 ± 0.12	0.12 ± 0.08	0.06 ± 0.03	0.00 ± 0.00
6 h	0.38 ± 0.11	0.23 ± 0.17	0.16 ± 0.05	0.08 ± 0.06
12 h	1.15 ± 0.41	0.58 ± 0.15	0.41 ± 0.18	0.03 ± 0.01
24 h	0.27 ± 0.07	0.20 ± 0.09	0.16 ± 0.07	0.07 ± 0.02
48 h	0.12 ± 0.04	0.07 ± 0.02	0.06 ± 0.01	0.04 ± 0.01
72 h	0.05 ± 0.02	0.05 ± 0.02	0.04 ± 0.01	0.05 ± 0.01

Even for the 25 nm particle, which have the fastest uptake transport rate among different sized micro-carriers, it took 12 h to reach the maximum accumulative amount ($1.15 \pm 0.41\%$ D) in visceral organs. Therefore, the estimated absorption transport rate of our micro-carrier with different size is below $0.1\% \text{ D h}^{-1}$ ($\frac{1.15 \pm 0.41\% \text{ D}}{12 \text{ h}}$).

3. There is very little discussion of the data, short-comings or comparisons with current or competing methods in development. In addition to these items, please discuss possible sources of toxicity of these microcarriers, transfer from the intestine to the blood, effects of microparticle size and considerations for using different proteins, etc.

Response: Thanks for the comments. Since there are a lot of questions in the comment 3, we will response them one by one.

Question (1): There is very little discussion of the data, short-comings or comparisons with current or competing methods in development.

Response (1): In order to discuss the data, short-comings and compare with current competing methods, we added the following contents in the revised manuscript:

“The ACIE method reported here is a NIR bioimaging technique which is based on the multi-excitation properties of the NIR-II DCNPs (Supplementary Fig. 21). Compared with the traditional radioisotopic method, the ACIE NIR bioimaging technique is non-invasive for *in vivo* detection. But only in-situ drug release information can be reflected by the ACIE. Other techniques such as radioisotopic method are still needed to track the trace and activity of the released drugs. To sum up, we expect ACIE technique would be strongly complementary to the traditional radioisotopic tracing method to realize non-invasive *in vivo* semi-quantitative monitoring of drug release.” (line 370, page 13.)

Question (2): In addition to these items, please discuss possible sources of toxicity of these microcarriers, transfer from the intestine to the blood, effects of microparticle size and considerations for using different proteins, etc.

Response (2): We have added following contents to discuss possible sources of toxicity of these micro-carriers and transfer route from the intestine to the blood on line 356, page 12:

“It is noted that the smaller micro-carriers, such as 25 nm micro-carriers, showed a relative short retention time (less than 24 h) and high uptake amount ($\sim 1.15\%$ at 12 h, Fig. 4b-g). Therefore, further toxicity investigation for the small micro-carriers is required before using, because these micro-carriers in the bloodstream may induce hemolysis, allergic reaction and even some severe diseases. In contrast, 1000 nm micro-carriers used in our protein delivery experiment exhibit many merits as a good oral drug carrier, such as extended retention time up to 72 h and little deposition ($< 0.1\%$) in the visceral organs. Long retention behavior of the drug carrier can be exploited as prolonged drug delivery system, which has more flexibility in dosage design than the conversional drug delivery system. And we also noticed that mice in our experiments did not show any signs of discomfort and their weight gain was normal (average weight increase $\sim 200\%$ after three months). Meanwhile, after seven days of daily receiving micro-carriers, histopathological examination showed that the morphology of intestinal tissue and tight junctions were intact without any signs of

degeneration (Fig. 5e-f), further indicating little healthy risky was caused by the micro-carriers.”

We have added following contents to discuss effects of microparticle size and considerations for using different proteins:

“Micro-carriers reported here provide many attractive features for oral drug delivery, such as high loading capacity, effective protection and high therapeutic index. Moreover, the properties of micro-carriers such as tunable pore size (2.5 - 12.0 nm, Supplementary Fig. 18, 19), aspect ratio (2 : 1 - 10 : 1, Supplementary Fig. 20), surface charge (negative or positive) and surface hydrophobicity endow them great potential in the field of drug delivery.” (line 340, page 12)

“In order to load a desired protein drug, there are several criteria for the design of micro-carrier: (1) pore size should be bigger than the desired protein drug; (2) the aspect ratio should be adequate to entrap the whole protein molecule; (3) micro-carrier and desired protein drug should carrying the opposite surface charge when loading.” (line 344, page 12)

Detailed comments:

4. According to Fig. 4a, at 0.25h, the NIR signal in 1000 nm particles group is higher than the other groups, especially 25nm group. It seems like the original amounts of DCNPs in each groups are different. Thus, the larger the particles, the more the DCNPs will be in the particles. The conclusion that larger particles can stay longer in GI tract than smaller particles could not be made without standardizing NIR signal. It will be better to standardize the NIR signal in each groups with relative DCNPs amount.

Response: Thanks for the useful comment. We have already standardized these NIR signals to make sure relative DCNPs amount is the same in each group (Supplementary Fig. 9). Therefore, the NIR signal intensity in Fig. 4a after standardizing only reflect the amount of micro-carriers in the GI tract in real time.

5. In Fig. 5g, it will be better to choose PEP loaded micro-carriers without SSPI covered + probe to be control. This could prove it is because SSPI that retain PEP in micro-carriers until releasing in intestine. This control could prove the higher FL intensity in PEP loaded micro-carriers+probe group is not because micro-carriers itself.

Response: Thanks for this comment. We have added PEP loaded micro-carriers without SSPI covered + probe as control. As shown in Supplementary Fig. 17, this control only behavior 2.5 times increase of the FL intensity, indicating most of PEP was deactivated in the stomach, further confirming the protection provided by SSPI is necessary in the micro-carrier drug delivery system.

In the revised manuscript, we have added the following contents on line 332, page 12: “In comparison, control groups including native PEP and micro-carriers without SSPI coating showed little activity because of the PEP degradation in the GI tract.”

We have added Supplementary Fig. 17 with the following figure caption: “*In vivo* bioimaging of mice after gavaged with peptide probe and PEP-NPTAT loaded micro-carriers without SSPI coating. 405 nm was used as excitation source. Representative images for n = 3 per group.”

6. What is the size of the pores in the microparticles that allow it to load protein drug? Can these

sizes be tunes or will pore size limit the choice of protein drug to load? What are the expected effects of aspect ratio, surface charge and surface hydrophobicity? (for instance, an antibody has a hydrodynamic radius of ~ 10 nm). K_a and k_q and equations (1-4) should be explained more clearly.

Response: Thanks for the comments. Since there are a lot of questions in the comment 6, we will response them one by one.

Question (1): *What is the size of the pores in the microparticles that allow it to load protein drug? Can these sizes be tunes or will pore size limit the choice of protein drug to load?*

Response (1): In a previous work reported by us (Shen, D., *et al.*, *Nano Lett.* **14**, 923-932 (2014).), we have discussed whether the different pore sizes can affect the loading and releasing amounts of protein drugs in detail. With bovine β -lactoglobulin (~ 5 nm) as a model protein, it was found that the micro-carriers with 5.5 nm pore size have the protein loading capacity of ~ 30.5 wt. %, while the micro-carriers with a larger mesopore size of 10 nm show a double loading capacity as 62.1 wt. %. Therefore, we can conclude that the pore size determines the loading capacity.

The pore size on the micro-carriers can be easily tuned by changing the reaction condition. In the revised manuscript, we have added the results of micro-carriers with different pore sizes (2.5, 7.7, 8.9 and 12.0 nm, **Supplementary Fig. 18, 19**). Therefore, the tunable pore sizes on the micro-carriers will allow a lot of protein drug available to load.

We have added the following figures in the revised manuscript:

“Supplementary Fig. 18. TEM images of micro-carriers with pore size of 2.5 nm (a), 7.7 nm (b), 8.9 nm (c) and 12.0 nm (d). The pore size can be easily tuned by changing the reaction condition, such as changing the oil used in the synthesis to octadecene (a), decahydronaphthalene (b) and cyclohexane (c). Micro-carriers with pore size of 12 nm (d) were synthesized by reducing the TEOS amount by 25% compared with (c).”

“Supplementary Fig. 19. Nitrogen adsorption-desorption isotherms (a) and pore size distribution (b) of different micro-carriers, respectively. The corresponding BET surface area was measured to be 586.4, 467.3, 366.8 and 330.1 $\text{m}^2 \text{g}^{-1}$, respectively.”

Question (2): *What are the expected effects of aspect ratio, surface charge and surface hydrophobicity?*

Response (2): The aspect ratio is directly related to the loading capacity and release rate of the protein drugs. Obviously, higher aspect ratio will result in higher loading capacity, but lower release rate. On the other hand, the loading capability of proteins will be limited if the aspect ratio is too low.

The surface charge and hydrophobicity of the micro-carriers should be carefully designed according to the charge and hydrophobicity of protein drugs. Generally, micro-carriers and protein drug are expected to have the opposite surface charge but same hydrophobicity when loading, because electrostatic and hydrophobic interaction can provide the micro-carriers a driving force to load the guest proteins. In contrast, same charges but opposite hydrophobicity is required for drug release. In our case, BSA-NPTAT composite was highly negative charged at pH 5, while the micro-carriers were positive charge due to the protonation of amino group on their surface (supplementary Fig. 1), the electrostatic interaction between them increased loading efficiency as well as the loading

rate. On the other hand, the micro-carriers will change to negative charge at pH 8, result in electrostatic repulsive interaction with the BSA-NPTAT composite. The influence of hydrophobicity change on the loading and releasing of protein drugs is caused by SSPI, which show controlled hydrophobicity in different pH condition (Fig. 1a). This unique character is the key point of the controlled drug release behavior of our micro-carrier.

We have added the following figures in the revised manuscript:

“**Supplementary Fig. 20.** TEM images of micro-carriers with aspect ratio of 2 : 1 (a), 6 : 1 (b) and 10 : 1 (c).”

Question (3): *K_A and k_q and equations (1-4) should be explained more clearly.*

Response (3): *K_A* is the binding constant, which reflect the degree of interactions between the NPTAT and BSA, while *k_q* is the bimolecular quenching constant between them. In order to make the explanation for *K_A*, *k_q* and eq (3-5) (equal to eq (2-4) in the original version) clearer for readers, we have added the following contents in the revised manuscript on line 167, page 6:

“Interaction between phthalocyanines (including NPTAT) and proteins has been widely investigated using FL spectroscopy method. By evaluating the FL quenching effect of proteins after adding a certain amount of NPTAT, the binding constant (*K_A*) and bimolecular quenching constant (*k_q*) between them can be quantified by the Stern-Volmer equation (see methods section, Eq. (3-5)). *K_A* reflects the degree of interaction between the NPTAT and proteins, while *k_q* can be used to identify the type of quenching process (static or dynamic quenching).”

And we also added the following contents to explain eq (1) in the revised manuscript on line 250, page 9:

“It is clearly demonstrated that the intensity of NIR-II signals excited by 808-nm laser were related to the quantity of the micro-carriers in real-time while those excited by 730-nm laser reflected the drug release percentage. Therefore, a semi-quantitative analysis is conducted to demonstrate the drug release percentage (*Q_(τ)*) of micro-carriers in real-time:

$$Q_{(\tau)} = \frac{F_{730,(\tau)} - F_{730,0}}{F_{730,unloaded} - F_{730,0}} \times 100\% = \frac{F_{730,(\tau)} - F_{730,0}}{\alpha F_{808,(\tau)} - F_{730,0}} \times 100\% \quad (1)$$

Where *F_{730,0}* is the initial intensity of NIR-II signals under 730-nm excitation of the drug loaded micro-carriers just after gavaging, and *F_{730,(τ)}* is used to represent the intensity of NIR-II signals of the drug loaded micro-carriers under 730-nm excitation at a specific time point (*τ*). *F_{730,unloaded}* is used to represent the intensity of NIR-II signals of the unloaded micro-carriers under 730 nm excitation. Since the intensity of NIR-II signals under 730-nm excitation will be quenched gradually with the increase of the drug loading amount, *F_{730,unloaded}* can be used to reflect the total amount of loaded BSA. However, it's difficult to obtain the *F_{730,unloaded}* in real-time because of the quenching effect of the loaded drug. But the *F_{730,unloaded}* can be related to *F_{808,(τ)}* directly, because the NIR-II signals from DCNPs under 730 nm and 808 nm excitation will maintain a fixed ratio when the power density is below 4.55 W cm⁻² (Supplementary Fig. 11, 12 and Supplementary Table 3). Therefore, *F_{730,unloaded}* can be expressed as *αF_{808,(τ)}*, where coefficient *α* is the constant to represent the *F_{730,unloaded}/F_{808,(τ)}*, which is measured to be 0.62 (see supplementary information). Furthermore, since the micro-carrier will be cleaned out of body with time, *F_{808,(τ)}* will change gradually during the drug delivery process. So *F_{808,(τ)}* can be used to reflect the amount of unloaded micro-carriers in real-time. Taking drug release amount calculation at 12 h after gavaging as an example, the intensity of NIR-II signals under 730 nm (*F_{730,(12 h)}*) or 808 nm (*F_{808,(12 h)}*) excitation was calculated to be 23.5

and 75.2, respectively (Fig. 6a). And the $F_{730,0}$ was measured to be 3.0, so the release percentage $Q_{(12\text{ h})}$ was estimated to be 47%. As shown in Fig. 6c, the calculated Q was only 10% at the first 2 h, then it showed a burst enhancement in the following 12 h and reached to ~ 62% at 72 h, suggesting the efficient release behavior of the micro-carriers *in vivo*.”

7. The use of mice versus rats should be clarified in the results (lines 207 and 228, 358)

Response: We appreciate this advice. We have changed “rats” to “mice” in the revised manuscript.

REVIEWERS' COMMENTS:

Reviewer #1 (Remarks to the Author):

Overall the manuscript has been revised based on prior comments. There are still language and grammatical errors that need to be corrected. For example,

Line 364. conversional drug delivery system

Line 369. healthy risky was caused by the micro-carriers

Reviewer #2 (Remarks to the Author):

To address the reviewers' comments, authors have undertaken a lot of more experiments. In the revised manuscript, authors have properly replied my critical questions including "why these particles cannot be absorbed by the body" and "why is the long retention time in GI tract helpful in oral drug delivery". I noticed that the 3rd reviewer also made similar comments. Great efforts have also been made to explain the pharmacokinetics of drug delivery, the surface effect for protein delivery, the mechanism that transfers particles from intestine into organs. After addressing the comments of reviewers, this interesting manuscript becomes much easier to follow. Given the originality of this approach and the fact that conclusions are adequately supported by experimental and blank experiments, I would like to recommend it for publication.

Reviewer #3 (Remarks to the Author):

After the revision, the researchers have provided solid data to support the feasibility of their non-invasive system to monitor drug release. Therefore, I recommend its publication on Nature Communications.

Major comments that have now been adequately addressed:

1. Clarify the goal of this system, including the potential mechanism of particle transport into the blood and organs.

The goal of this absorption competition induced emission (ACIE) bioimaging system is to deliver drugs to the intestine. The researchers have added the uptake percentage of different sized micro-carriers in internal organs to support the low uptake transport rate. The surface properties of the micro-carrier in the intestine is hydrophilic and negatively charged. These surface properties will hinder the absorption of the micro-carrier through Peyer's patches. It seems like whether the loaded drug will transfer from the intestine into the blood or organs is depended on the loaded drug itself, not depended on the micro-carrier.

2. Use a conventional method to validate the PK of the loaded protein.

The researchers have added conventional radioisotopic tracing method to validate the PK data of released protein. ¹²⁵I-BSA was loaded in micro-carriers. Few of released proteins were absorbed into the blood and internal organs. After 72 hours, 25% of released proteins were cleaned through the urine. The ACIE bioimaging technique developed by the researchers can only give the real time BSA release amount without the distribution of the released BSA. Thus, other traditional techniques are still needed to track the released drugs. ACIE bioimaging technique could be a complementary to the traditional tracing method.

3. Why is fluorophore not covalently to BSA? Proteins and NPTAT release rates should be quantified

and compared.

The researchers have added data to prove NPTAT can be used as a reliable tracer for loaded proteins. Previous studies have proved the feasibility of non-covalent interaction between phthalocyanines and proteins. In addition, covalent bind might induce deactivation of loaded proteins.

Detailed comments that have been addressed:

5. Standardize the NIR signal in each groups with relative DCNPs amounts.

The researcher have standardized the NIR signal with relative DCNPs amounts. The results validate that larger particles can stay longer in GI tract compared with smaller ones.

6. Choose PEP loaded micro-carriers without SSIP covered + probe as control.

Suitable control has already been added to exclude the possible interaction of micro-carriers itself.

7. About the size of the pores in the micro-carrier, effects of aspect ratio, surface charge and surface hydrophobicity. Explain K_a and k_q and equations.

The pore size on the micro-carrier can be tuned by changing the reaction condition. The researchers have added results of different pore sizes. The aspect ratio is related to the loading capacity and release rate of loaded drugs. Micro-carriers and proteins are expected to have the opposite surface charge but same hydrophobicity to provide a driving force to load proteins. SSPI can change the hydrophobicity in different pH conditions to control drug release of the micro-carrier. The explanation of surface properties is helpful to understand the mechanism of this novel bioimaging system. K_a and k_q and equation have been clearly explained.

8. Use of mice versus rats.

Use of mice has been unified in the revised manuscript.

Point-by-Point Response to Referees

Reviewers' comments:

Reviewer #1 (Remarks to the Author):

Overall the manuscript has been revised based on prior comments. There are still language and grammatical errors that need to be corrected. For example,

Line 364. conversional drug delivery system

Line 369. healthy risky was caused by the micro-carriers

Response: Thanks so much for the useful comments and suggestions. We have polished our manuscript carefully and corrected the grammatical, styling, and mistyped errors found in our manuscript. Detailed modifications are also provided as following. **Furthermore**, in order to make sure the high quality, this manuscript has been edited by Nature Publishing Group Language Editing Service before the publication.

We corrected the following mistake on page 13, line 377: “conversional drug delivery system” was changed to “conventional drug delivery system”.

We corrected the following mistake on page 13, line 382: “healthy risky was caused by the micro-carriers” was changed to “healthy risk was caused by the micro-carriers”.

Reviewer #2 (Remarks to the Author):

To address the reviewers' comments, authors have undertaken a lot of more experiments. In the revised manuscript, authors have properly replied my critical questions including “why these particles cannot be absorbed by the body” and “why is the long retention time in GI tract helpful in oral drug delivery”. I noticed that the 3rd reviewer also made similar comments. Great efforts have also been made to explain the pharmacokinetics of drug delivery, the surface effect for protein delivery, the mechanism that transfers particles from intestine into organs. After addressing the comments of reviewers, this interesting manuscript becomes much easier to follow. Given the originality of this approach and the fact that conclusions are adequately supported by experimental and blank experiments, I would like to recommend it for publication.

Response: We appreciate the reviewer for the positive comment.

Reviewer #3 (Remarks to the Author):

After the revision, the researchers have provided solid data to support the feasibility of their non-invasive system to monitor drug release. Therefore, I recommend its publication on Nature Communications.

Major comments that have now been adequately addressed:

1. Clarify the goal of this system, including the potential mechanism of particle transport into the blood and organs.

The goal of this absorption competition induced emission (ACIE) bioimaging system is to deliver drugs to the intestine. The researchers have added the uptake percentage of different sized micro-carriers in internal organs to support the low uptake transport rate. The surface properties of the micro-carrier in the intestine is hydrophilic and negatively charged. These surface properties will hinder the absorption of the micro-carrier through Peyer's patches. It seems like whether the loaded drug will transfer from the intestine into the blood or organs is depended on the loaded drug itself, not depended on the micro-carrier.

Response: Thanks so much for the comment. The goal of our micro-carrier system is to deliver drugs to the intestine, while the ACIE bioimaging technique can be utilized to monitor this delivery process.

2. Use a conventional method to validate the PK of the loaded protein.

The researchers have added conventional radioisotopic tracing method to validate the PK data of released protein. ¹²⁵I-BSA was loaded in micro-carriers. Few of released proteins were absorbed into the blood and internal organs. After 72 hours, 25% of released proteins were cleaned through the urine. The ACIE bioimaging technique developed by the researchers can only give the real time BSA release amount without the distribution of the released BSA. Thus, other traditional techniques are still needed to track the released drugs. ACIE bioimaging technique could be a complementary to the traditional tracing method.

Response: Thanks so much for the positive comment.

3. Why is fluorophore not covalently to BSA? Proteins and NPTAT release rates should be quantified and compared.

The researchers have added data to prove NPTAT can be used as a reliable tracer for loaded proteins. Previous studies have proved the feasibility of non-covalent interaction between phthalocyanines and proteins. In addition, covalent bind might induce deactivation of loaded proteins.

Response: Thanks for the positive comment.

Detailed comments that have been addressed:

5. Standardize the NIR signal in each groups with relative DCNPs amounts.

The researcher have standardized the NIR signal with relative DCNPs amounts. The results validate that larger particles can stay longer in GI tract compared with smaller ones.

Response: We appreciate this positive comment.

6. Choose PEP loaded micro-carriers without SSIP covered + probe as control.

Suitable control has already been added to exclude the possible interaction of micro-carriers

itself.

Response: Thanks for the positive comment.

7.About the size of the pores in the micro-carrier, effects of aspect ratio, surface charge and surface hydrophobicity. Explain K_a and k_q and equations.

The pore size on the micro-carrier can be tuned by changing the reaction condition. The researchers have added results of different pore sizes. The aspect ratio is related to the loading capacity and release rate of loaded drugs. Micro-carriers and proteins are expected to have the opposite surface charge but same hydrophobicity to provide a driving force to load proteins. SSPI can change the hydrophobicity in different pH conditions to control drug release of the micro-carrier. The explanation of surface properties is helpful to understand the mechanism of this novel bioimaging system. K_a and k_q and equation have been clearly explained.

Response: Thanks so much for the positive comment.

8. Use of mice versus rats.

Use of mice has been unified in the revised manuscript.

Response: We appreciate the positive comment.